# Sorting nexin-27 regulates AMPA receptor trafficking through the synaptic adhesion protein LRFN2

Kirsty J McMillan[1][†]*, Paul J Banks[2][†], Francesca LN Hellel[1], Ruth E Carmichael[1], Thomas Clairfeuille[3], Ashley J Evans[1], Kate J Heesom[4], Philip Lewis[4], Brett M Collins[3], Zafar I Bashir[2], Jeremy M Henley[1], Kevin A Wilkinson[1]*, Peter J Cullen[1]*

[1]School of Biochemistry, University of Bristol, Bristol, United Kingdom; [2]School of Physiology, Pharmacology and Neuroscience, University of Bristol, Bristol, United Kingdom; [3]Institute for Molecular Bioscience, The University of Queensland, Queensland, Australia; [4]Proteomics facility, School of Biochemistry, University of Bristol, Bristol, United Kingdom

**Abstract** The endosome-associated cargo adaptor sorting nexin-27 (SNX27) is linked to various neuropathologies through sorting of integral proteins to the synaptic surface, most notably AMPA receptors. To provide a broader view of SNX27-associated pathologies, we performed proteomics in rat primary neurons to identify SNX27-dependent cargoes, and identified proteins linked to excitotoxicity, epilepsy, intellectual disabilities, and working memory deficits. Focusing on the synaptic adhesion molecule LRFN2, we established that SNX27 binds to LRFN2 and regulates its endosomal sorting. Furthermore, LRFN2 associates with AMPA receptors and knockdown of LRFN2 results in decreased surface AMPA receptor expression, reduced synaptic activity, and attenuated hippocampal long-term potentiation. Overall, our study provides an additional mechanism by which SNX27 can control AMPA receptor-mediated synaptic transmission and plasticity indirectly through the sorting of LRFN2 and offers molecular insight into the perturbed function of SNX27 and LRFN2 in a range of neurological conditions.

*For correspondence:
kirsty.mcmillan@bristol.ac.uk (KJMM);
Kevin.Wilkinson@bristol.ac.uk (KAW);
Pete.Cullen@bristol.ac.uk (PJC)

[†]These authors contributed equally to this work

Competing interests: The authors declare that no competing interests exist.

## Introduction

The endosomal network plays a central role in controlling the functionality of the cell surface through orchestrating the sorting of endocytosed integral proteins between two fates: either degradation within the lysosome or retrieval from degradation for active recycling back to the plasma membrane (*Cullen and Steinberg, 2018*). While the molecular details of sorting to the degradative fate are relatively well described, only recently, with the identification of sorting nexins (*Kurten et al., 1996*; *Stockinger et al., 2002*; *Joubert et al., 2004*; *Carlton et al., 2004*; *Strochlic et al., 2007*; *Harterink et al., 2011*), retromer (*Seaman et al., 1998*), retriever (*McNally et al., 2017*), and the WASH, CCC and ESCPE-1 complexes (*Derivery et al., 2009*; *Gomez and Billadeau, 2009*; *Phillips-Krawczak et al., 2015*; *Simonetti et al., 2019*), have the core and evolutionarily conserved sorting complexes that orchestrate retrieval and recycling begun to be identified. Importantly, an increasing body of clinical evidence is linking mutations in these sorting complexes with a variety of human pathologies, most notably neurological diseases and disorders, metabolic conditions and pathogen infections (*Cullen and Steinberg, 2018*). With the notable exception of defects in the CCC complex-mediated retrieval and recycling of low-density lipoprotein (LDL) receptor and the clearance of circulating LDL-cholesterol during hypercholesterolaemia and atherosclerosis (*Bartuzi et al., 2016*; *Fedoseienko et al., 2018*), how defects in the cell surface proteome, that arise from perturbed

endosomal sorting, relate to the aetiology of these complex disorders remains poorly understood. A case in point is sorting nexin-27 (SNX27), in which destabilised expression is associated with Down's Syndrome and coding mutations are observed in patients with pleomorphic phenotypes that have at their core epilepsy, developmental delay and subcortical white matter abnormalities (*Damseh et al., 2015*; *Parente et al., 2020*; *Wang et al., 2013*).

SNX27 is unique within the sorting nexin family in that it contains an amino-terminal PSD-95, Disclarge and ZO-1 (PDZ) domain. This serves a bifunctional role mediating two mutually exclusive protein:protein interactions: first, the binding to the heterotrimeric retromer complex (*Gallon et al., 2014*; *Steinberg et al., 2013*); and secondly, the binding to a specific type of PDZ domain-binding motif located at the carboxy-terminus of an array of integral proteins (*Cullen, 2008*; *Steinberg et al., 2013*; *Lauffer et al., 2010*; *Temkin et al., 2011*). Through these interactions SNX27 regulates the retromer-dependent retrieval of internalised integral proteins that contain the specific PDZ domain-binding motif and promotes their recycling to the plasma membrane (*Lauffer et al., 2010*; *Temkin et al., 2011*; *Steinberg et al., 2013*).

The identification and functional validation of a handful of neuronal integral proteins that rely on SNX27 for their trafficking has provided some insight into the complex aetiology of SNX27-associated pathology. These proteins include α-amino-3-hydroxy-5-methyl-4-isoxazolepropionic acid (AMPA) receptors (*Wang et al., 2013*; *Hussain et al., 2014*; *Loo et al., 2014*), N-methyl-D-aspartate (NMDA) receptors (*Cai et al., 2011*), serotonin (5-HT) receptors (*Joubert et al., 2004*) and the G protein–activated inward rectifying potassium channels (GIRK/Kir3) (*Lunn et al., 2007*). From studies in SNX27 knockout mice, it is clear that SNX27 is required to maintain AMPA-receptor-mediated postsynaptic currents during the process of synaptic plasticity (*Wang et al., 2013*; *Loo et al., 2014*; *Cai et al., 2011*). Indeed, the de-regulation of SNX27 expression associated with Down's syndrome (*Wang et al., 2013*), is considered to lead to synaptic dysfunction, in part, through reduced SNX27-mediated AMPA receptor trafficking. SNX27 therefore plays a pivotal role in the endosomal sorting of AMPA receptors. The current dogma describing SNX27-mediated AMPA receptor sorting relies upon the direct binding of the PDZ binding motif presented at the carboxy-termini of AMPA receptor subunits to the PDZ domain of SNX27. However, using isothermal titration calorimetry Clairfeuille and colleagues were unable to detect quantifiable binding of SNX27 to the AMPA receptor subunits GluA1 and GluA2. Moreover, the authors failed to observe binding upon GluA1 phosphorylation, or upon association of SNX27 with the retromer component VPS26, which is known to enhance SNX27 binding to other PDZ binding motif-containing cargoes (*Clairfeuille et al., 2016*). These data suggest a need to reflect on the molecular details of SNX27-mediated endosomal sorting of AMPA receptors.

Here, using unbiased proteomics, we define the SNX27 interactome in primary rat cortical neurons and describe the identification of new SNX27-dependent neuronal cargo. Many of these integral proteins provide further molecular insight into the underlying neuronal de-regulation associated with SNX27-associated pathologies. In particular, we functionally validate one specific cargo, the synaptic adhesion molecule leucine-rich repeat and fibronectin type-III domain containing protein 2 (LRFN2). We identify that SNX27 directly associates with the carboxy-terminal PDZ binding motif of LRFN2, while the amino-terminal region of LRFN2 associates with AMPA receptors. Functionally, SNX27 is required for the cell surface recycling of endocytosed LRFN2 and AMPA receptors. By establishing that LRFN2 knockdown also results in decreased surface expression of AMPA receptors and, in ex vivo recordings, the reduction of synaptic activity and attenuation of hippocampal long-term potentiation, we add to the known complexities of SNX27-mediated AMPA receptor endosomal sorting by proposing a role for LRFN2 in bridging the indirect association of SNX27 with AMPA receptors.

## Results

### A neuronal SNX27 interactome reveals new cargoes for SNX27-mediated trafficking

To identify neuronal cargoes that depend on SNX27 for their trafficking, we took an unbiased proteomic approach to quantify the SNX27 interactome in primary rat cortical neuronal cultures. Here, we transduced cortical neurons with sindbis virus expressing either GFP or GFP-SNX27 and verified that

the GFP-SNX27 retained the ability to localise to endosomes as defined by co-localisation with the early endosomal marker EEA1 (*Figure 1A*). After 24 hr, we performed GFP-nanotrap immunoisolation followed by protein digestion and tandem mass tagging (TMT) of the resulting peptides. Interactors were identified quantitatively using liquid chromatography-tandem mass spectrometry and quantified across three independent biological repeats. A single list of SNX27-interacting proteins (276 proteins) was initially resolved by excluding proteins not present in all three datasets (*Figure 1B* and *Supplementary file 1*). We further refined these data by excluding proteins that had an average log-fold change of GFP-SNX27:GFP of less than two and removed identified proteins if they did not meet statistical significance (p<0.05). The resulting 212 proteins were considered to comprise a high confidence cortical neuronal SNX27 interactome (*Figure 1C* and *Supplementary file 1*).

Confirming the validity of our approach, many established SNX27 interactors were identified including the retromer and WASH complexes (*Steinberg et al., 2013*) and OTU deubiquitinase with linear linkage specificity (OTULIN) – a deubiquitinase with specificity for Met1-linked ubiquitin chains (*Stangl et al., 2019*). We used gene ontology analysis (PANTHER Classification System; p < 0.05) to assess the neuronal SNX27 interactome and found that many of the SNX27 interactors were classified in having a role in neuronal development and differentiation, as well as intracellular transport (*Figure 1D*). Out of the 212 identified proteins sixteen contained a Type I PDZ binding motif with the optimal acidic amino acid at the $-3$ position required for high-affinity binding to SNX27 (*Clairfeuille et al., 2016*). Within this cohort, seven proteins were classified as integral proteins (shown in orange in *Figure 1C* and *Supplementary file 1*): the high-affinity glutamate transporter, SLC1A3 (*Storck et al., 1992*) (PDZ binding motif – $E^{-3}$-T-K-$M^{-0}$) and the sodium bicarbonate co-transporter, SLC4A7 (*Thornell and Bevensee, 2015*) ($E^{-3}$-T-S-$L^{-0}$); a sodium-dependent transporter, SLC6A11 (*Borden, 1996*) ($E^{-3}$-T-H-$F^{-0}$); an outward rectifying potassium channel, KCNT2 (*Bhattacharjee et al., 2003*) ($E^{-3}$-T-Q-$L^{-0}$); a scaffold in neurotrophin signalling, KIDINS220 (*Iglesias et al., 2000*; *Kong et al., 2001*) ($E^{-3}$-S-I-$L^{-0}$); a receptor for the neuronal secreted protein LGI1 (*Fukata, 2006*), ADAM22 ($E^{-3}$-T-S-$I^{-0}$), and LRFN2 ($E^{-3}$-S-T-$V^{-0}$), a protein previously implicated in the synaptic clustering of glutamate receptors (*Ko et al., 2006*; *Morimura et al., 2006*; *Wang et al., 2006*), and genetically associated with patients harbouring working memory deficits (*Thevenon et al., 2016*).

Interestingly, we failed to classify any AMPA receptor subunits as components of the neuronal SNX27 interactome. That said, GluA2 was quantified in one data set but did not fulfil the stringent filtering criteria and hence was not annotated in the final interactome. This could reveal a weak, low abundance association between SNX27 and GluA2, which may reflect an indirect mechanism of association. However, to independently examine the binding of SNX27 to the PDZ-binding motifs of AMPA receptor subunits, we generated a series of amino-terminal GFP-tagged AMPA receptor fusion proteins by cloning the carboxy-terminal tails of rat GluA1, GluA2, GluA3, and GluA4 into a mammalian GFP expression vector. To act as positive controls, we also cloned the carboxy-terminal tails of the high-affinity glutamate transporter SLC1A3 and the sodium bicarbonate co-transporter SLC4A7. These two proteins contain an optimal motif for high-affinity binding to the SNX27 PDZ domain as defined by an acidic residue at the $-3$ position of their Type I PDZ binding motif (*Clairfeuille et al., 2016*; *Figure 1E*). The AMPA receptor PDZ binding motifs lack this optimal sequence. Alongside the wild-type tails of SLC1A3 and SLC4A7, we also generated corresponding PDZ binding motif mutants through removal of the last three carboxy-terminal amino acids. The resulting series of plasmids were transiently transfected into human embryonic kidney (HEK293T) cells prior to GFP-nanotrap immunoisolation and quantitative western blotting of the resulting precipitates. While both GFP-SLC1A3 and GFP-SLC4A7 efficiently pulled down endogenous SNX27, in a manner dependent on their PDZ binding motifs (*Figure 1F*), we failed to observe detectable association of endogenous SNX27 with any of the GFP-tagged AMPA receptor subunits (*Figure 1G*).

To ensure that the lack of detectable binding did not arise from species cross reactivity, an unlikely event given the high-sequence conservation between rat and human SNX27, we also performed a series of co-immunoprecipitation experiments in HEK293T cells expressing the GFP-tagged AMPA receptor carboxy-terminal tails and Flag-tagged full-length rat SNX27. Again, we failed to observe a detectable association of AMPA receptor subunits with rat SNX27 (*Figure 1—figure supplement 1*). Finally, to examine whether SNX27 could potentially interact with AMPA receptors specifically in the context of neurons we transduced primary rat cortical neuronal cultures with

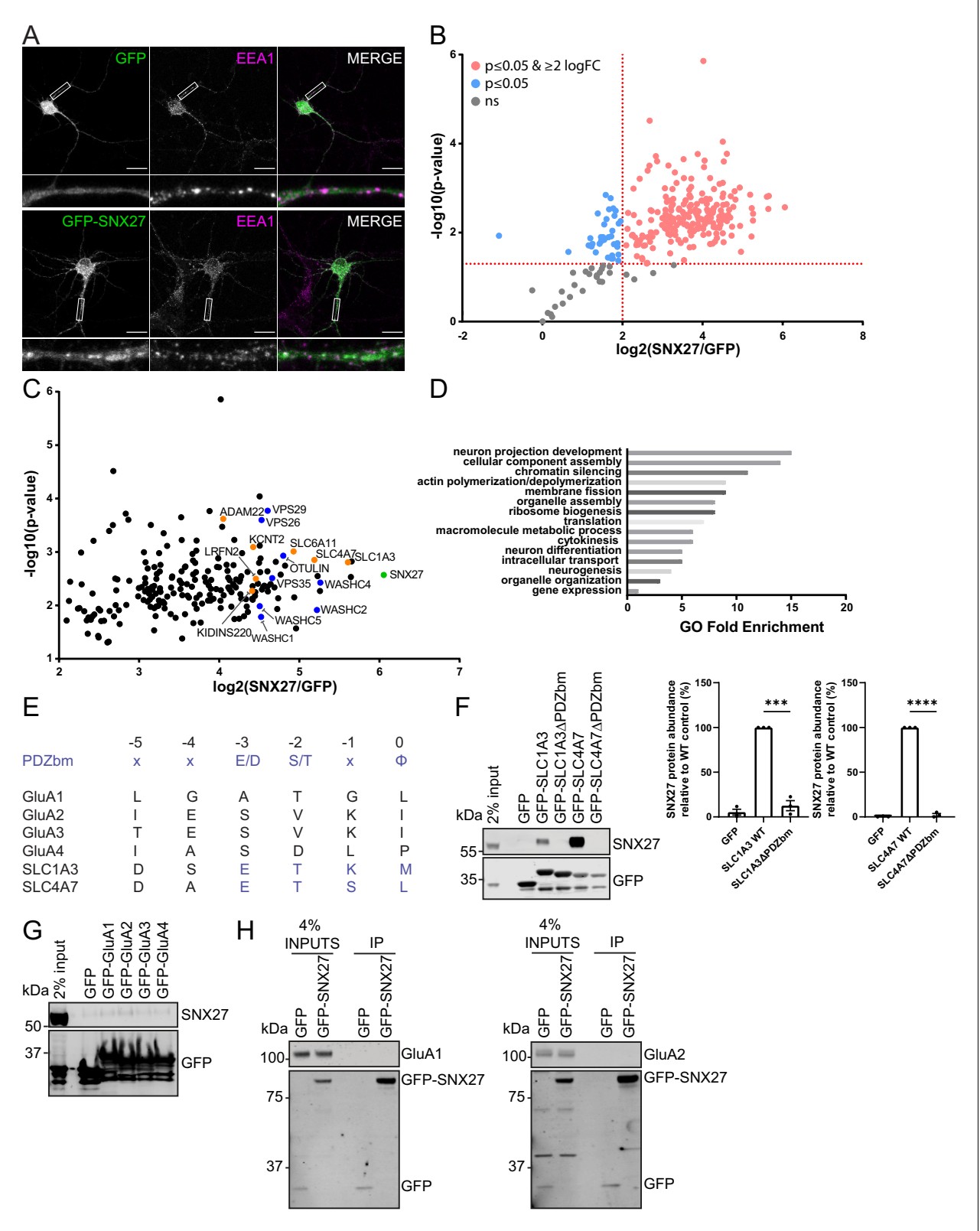

**Figure 1.** SNX27 interactome reveals new neuronal cargoes. (**A**) Immunofluorescence staining of endogenous early endosomal marker EEA1 (pseudocoloured in magenta) in DIV21 rat cortical neurons transduced with GFP or GFP-SNX27 expressing sindbis virus. Scale bars, 20 μm. White boxes indicate the 20 μm section zoomed in. (**B**) TMT Interactome of SNX27 compared to the GFP control quantified across three independent experiments (n = 3) in DIV21 rat cortical neurons. Plotted proteins were present in all three data sets and analysed using a one-sample t-test and Benjamini–

*Figure 1 continued on next page*

*Figure 1 continued*

Hochberg false-discovery rate (276 proteins). Vertical red line represents the threshold for enriched proteins in the SNX27 interactome ($\geq 2$) compared to the GFP control. Horizontal red line represents the threshold for statistical analysis ($p \leq 0.05$). Pink circles are the protein interactors that are enriched and meet statistical significance. Blue circles are the proteins that meet statistical significance but are not over twofold enriched. Grey circles are interactions that are not statistically significant. (C) Filtered TMT interactome (212 proteins) showing only those proteins that were statistically significant and over two log fold change. SNX27 is shown in green with retromer, and the WASH complex shown in blue. Transmembrane proteins that contain a Type I PDZbm with the acidic residue at the −3 position are shown in orange. (D) Gene ontology analysis using the PANTHER classification system of the filtered SNX27 interactome. (E) Schematic highlighting the last six amino acids of the C-terminal tails of isoform 1 of human GluA1, GluA2, GluA3, GluA4, SLC1A3, and SLC4A7 (sequences are conserved in rat). Only SLC1A3 and SLC4A7 possess the optimal PDZ binding motif (PDZbm) consensus sequence for high-affinity binding to SNX27 (highlighted in blue, φ; hydrophobic amino acid). (F–H) Fluorescence-based western analysis after GFP-Trap immunoprecipitation of the: (F) C-terminal tails of GFP-SLC1A3 and GFP-SLC4A7 (+/- PDZbm) with endogenous SNX27 in HEK293T cells. Quantification from three independent experiments (n = 3). Data expressed as a percentage of the WT condition and analysed by an unpaired t-test. Error bars represent mean ± SEM. ****, $p \leq 0.0001$; ***, $p \leq 0.001$. (G) C-terminal tails of AMPA receptors (GFP-GluA1-4) with endogenous SNX27 in HEK293T cells. (H) Full-length GFP-SNX27 expressing sindbis virus with endogenous GluA1 or GluA2 in DIV20 rat cortical neurons.

The online version of this article includes the following source data and figure supplement(s) for figure 1:

Source data 1. Original immunoblots.
Source data 2. Data for *Figure 1F*.
Figure supplement 1. SNX27 does not directly interact with AMPA receptors.
Figure supplement 1—source data 1. Original immunoblots.

sindbis virus expressing either GFP or GFP-SNX27. Twenty-four hr after transduction we carried out GFP-nanotrap immunoisolation and quantitative western blotting for the endogenous AMPA receptor subunits GluA1 and GluA2. Again, under these conditions, we failed to detect any association of endogenous GluA1 or GluA2 with rat SNX27 (*Figure 1H*). Together these data are consistent with the AMPA receptor PDZ binding motifs lacking the optimal sequence for high-affinity SNX27 binding (*Clairfeuille et al., 2016*) and corroborate our proteomic data. Whilst our data do not exclude that SNX27 may directly bind to AMPA receptors under specific circumstances, our data is more consistent with the SNX27-dependent recycling of AMPA receptors being principally controlled by indirect binding. Hence, we assessed the SNX27 interactome further, to identify any interactors that may serve to 'bridge' the association of SNX27 with AMPA receptors. Of particular interest in this regard was LRFN2, a protein known to be involved in synaptic clustering (*Ko et al., 2006*; *Morimura et al., 2006*; *Wang et al., 2006*).

## LRFNs interact with SNX27 through their PDZ-binding motifs

The LRFN family (also known as synaptic adhesion-like molecules [SALMs]) comprises five single transmembrane spanning proteins, LRFN1 through to LRFN5, that each contain an extracellular region of six leucine-rich repeats (LRR), an immunoglobulin (Ig) domain and a fibronectin type III domain, but differ in their cytosolic facing carboxy-terminal tails with LRFN1, LRFN2, and LRFN4 containing a Type I PDZ binding motif (*Lie et al., 2018*; *Seabold et al., 2008*). Both LRFN1 and LRFN4 were identified in the raw SNX27 proteomic data sets but were each filtered out from the final high confidence interactome because they were not quantified across all three biological repeats (*Supplementary file 1*). To validate the association of SNX27 with LRFN2, we cloned the carboxy-terminal cytoplasmic tails of all five LRFNs into mammalian GFP expression vectors. Transient transfection into HEK293T cells followed by GFP-nanotrap immunoisolation of the GFP-LRFN fusion proteins and quantitative western blotting revealed that only those LRFNs containing a PDZ binding motif, LRFN1, LRFN2 and LRFN4, associated with endogenous SNX27 (*Figure 2A*). Moreover, each association was dependent on the presence of the corresponding PDZ binding motif as deletion of the last three amino acids resulted in mutant LRFNs that failed to associate with SNX27 (*Figure 2B*).

The LRFN2 PDZ binding motif is defined by the sequence $E^{-3}$-$S^{-2}$-$T^{-1}$-$V^{0}$, where the valine residue constitutes the carboxy-terminal amino acid. Mutation of the glutamic acid residue at the −3 position (LRFN2 (p.E786A)) or the carboxy-terminal valine (LRFN2 (p.V789A)) lead to the formation of mutant proteins that each lost the ability to associate with SNX27 (*Figure 2C*). To define the direct nature of the SNX27 PDZ domain binding to the LRFN2 PDZ binding motif, we turned to isothermal titration calorimetry (ITC). This established that the isolated recombinant PDZ domain of

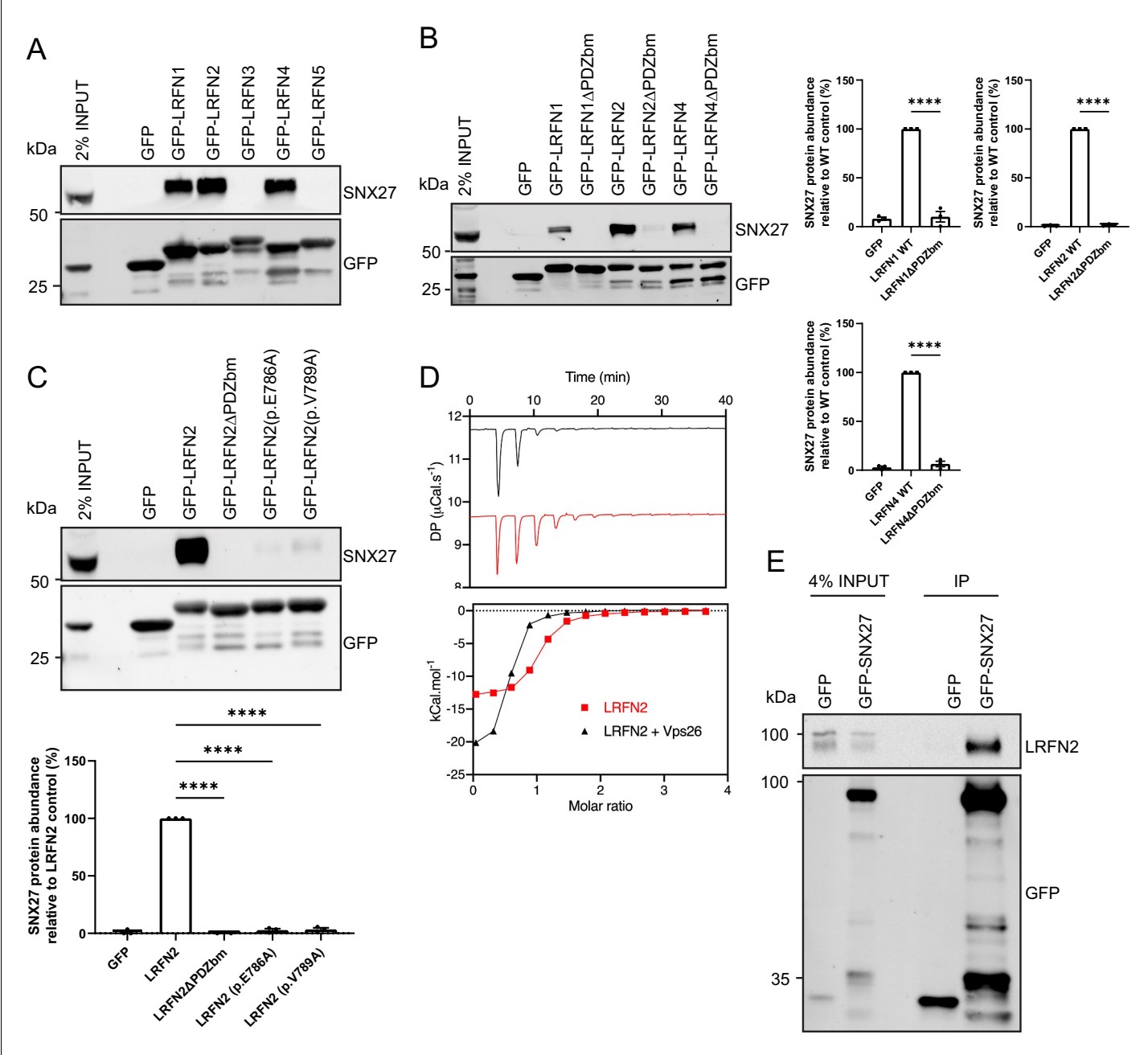

**Figure 2.** LRFNs interact with SNX27 through their PDZ binding motifs. (A–C) Fluorescence-based western analysis after GFP-Trap immunoprecipitation of: (A) the C-terminal tails of GFP-LRFN1-5 with endogenous SNX27 in HEK293T cells; (B) the C-terminal domains of LRFN1, LRFN2, and LRFN4 (+/- PDZ binding motif (PDZbm)) with endogenous SNX27 in HEK293T cells. Quantification from three independent experiments (n = 3). Data expressed as a percentage of the wild-type condition and analysed by an unpaired t-test; (C) the C-terminal domain of LRFN2 (+/- PDZbm and mutants pE786A, pV789A) with endogenous SNX27 in HEK293T cells. Quantification from three independent experiments (n = 3). Data expressed as a percentage of the wild-type condition and analysed by an unpaired t-test. (D) Binding of the LRFN2 peptide to the SNX27 PDZ domain, measured by ITC either to SNX27 alone or in the presence of the retromer component VPS26. Top panel shows raw data and bottom panel shows integrated and normalised data. (E) Fluorescence-based western analysis after GFP-Trap immunoprecipitation of full-length GFP-SNX27 with endogenous LRFN2 in DIV20 rat cortical neurons. In all figures error bars represent mean ± SEM. ****, $p \leq 0.0001$.

The online version of this article includes the following source data for figure 2:

**Source data 1.** Original immunoblots.
**Source data 2.** Data for *Figure 2B and C*.

SNX27 directly bound to a synthetic peptide corresponding to the LRFN2 PDZ binding motif, S-S-E-W-V-M-E$^{-3}$-S-T-V$^{-0}$ with a high micromolar affinity ($K_d$ = 1.6 µM) (*Figure 2D*). Moreover, the affinity of this interaction was enhanced upon inclusion of recombinant VPS26 ($K_d$ < 1.0 µM), a retromer component that directly associates with the PDZ domain of SNX27 and has been previously shown to enhance the binding affinity between the SNX27 PDZ domain and PDZ binding motif-containing peptides (*Gallon et al., 2014*; *Chan et al., 2016*).

Finally, to confirm that endogenous LRFN2 also associated with SNX27, we transduced primary rat cortical neuronal cultures with sindbis virus expressing either GFP or GFP-SNX27. GFP-nanotrap immunoisolation and quantitative western blotting confirmed the association between SNX27 and full-length endogenous LRFN2 (*Figure 2E*). Together these data establish that by means of its carboxy-terminal PDZ binding motif LRFN2 directly associates with the PDZ domain of SNX27. We also suggest that this mode of interaction holds true for LRFN1 and LRFN4.

## The membrane trafficking of LRFN2 is dependent on SNX27

To examine the functional importance of SNX27 binding to LRFN2, we transduced primary rat hippocampal neuronal cultures with sindbis virus encoding for GFP-tagged SNX27 and mCherry-tagged full-length LRFN2 (we were unable to identify antibodies suitable for detecting the expression of endogenous SNX27 or endogenous LRFN2 by immunocytochemistry in these primary cultures). Confocal microscopy revealed that the endosome associated SNX27 (see *Figure 1A*) co-localised with LRFN2 punctae throughout the neuron including in the dendrites and the cell body (*Figure 3A*). To determine the role of SNX27 in the steady-state localisation of LRFN2, we transduced primary rat cortical neuronal cultures with SNX27 shRNA or a non-targeting control shRNA (*Binda et al., 2019*). Following 7 days of incubation, during which time the expression of endogenous SNX27 was strongly suppressed (*Figure 3B*), we biochemically quantified the total cellular levels of endogenous LRFN2 by western analysis. Similar to a wide array of other integral proteins that require endosomal SNX27 for their retrieval away from the lysosomal degradative fate (*Steinberg et al., 2013*), the suppression of SNX27 expression led to a robust reduction in the total cellular level of LRFN2 (an approximate 50% reduction, n = 5, unpaired t-test, t (8) = 4.6, p = 0.0017) (*Figure 3C*). We also performed restricted cell surface biotinylation and streptavidin affinity capture coupled with quantitative western analysis to quantify the cell surface level of the AMPA receptor subunits, GluA1 and GluA2, and LRFN2. Consistent with published studies (*Hussain et al., 2014*; *Loo et al., 2014*; *Wang et al., 2013*), the suppression of SNX27 expression led to a clear reduction in the cell surface level of GluA1 and GluA2 (an approximate 37% reduction for GluA1, n = 4, unpaired t-test, t (6) = 3.9, p = 0.0078; and an approximate 64% reduction for GluA2, n = 4, unpaired t-test, t (6) = 4.0, p = 0.0074) (*Figure 3D*). Transferrin receptor levels were also measured as a control and were found to be unchanged (n = 4, unpaired t-test, t (6) = 0.6, p = 0.5774) (*Figure 3E*). Importantly, under these conditions, SNX27 suppression also induced a robust reduction in the cell surface level of LRFN2 (an approximate 52% reduction, n = 6, unpaired t-test, t (10) = 5.0, p = 0.0006) (*Figure 3F*).

To assess further the trafficking of LRFN2, we analysed the internalisation of LRFN2 after suppression of SNX27 in the neuroglioma H4 cell line. After 6 days of culturing with either a control or SNX27 shRNA, the cells were transfected with mCherry-LRFN2 and left for another 24 hr. Surface LRFN2 was labelled using an mCherry antibody, to detect the exofacial expressed mCherry of the LRFN2 fusion protein, and then allowed to internalise for 2 hr. After fixation and permeabilisation, the cells were co-stained with LAMP2 to label lysosomes. To detect the mCherry antibody, we used an Alexa Fluor 405 conjugated secondary antibody. The colocalisation between LRFN2 and LAMP2 significantly increased after SNX27 suppression (an approximate 27% increase, n = 30, unpaired t-test, t (58) = 2.4, p = 0.0200) (*Figure 3G*). Together these data establish LRFN2 as an integral protein that conforms to the dogma of SNX27-mediated membrane trafficking in that it requires SNX27 for its retrieval from lysosomal degradation, a pre-requisite for recycling back to the cell surface.

## LRFNs interact with AMPA receptors

The LRFN family of synaptic proteins are considered adhesion molecules that cluster receptors at the synaptic surface (*Ko et al., 2006*; *Morimura et al., 2006*; *Wang et al., 2006*; *Lie et al., 2018*). For LRFN2, research has principally focused on its ability to cluster NMDA receptors (*Wang et al., 2006*). To define whether LRFN2 also plays a role in regulating AMPA receptor trafficking, we first

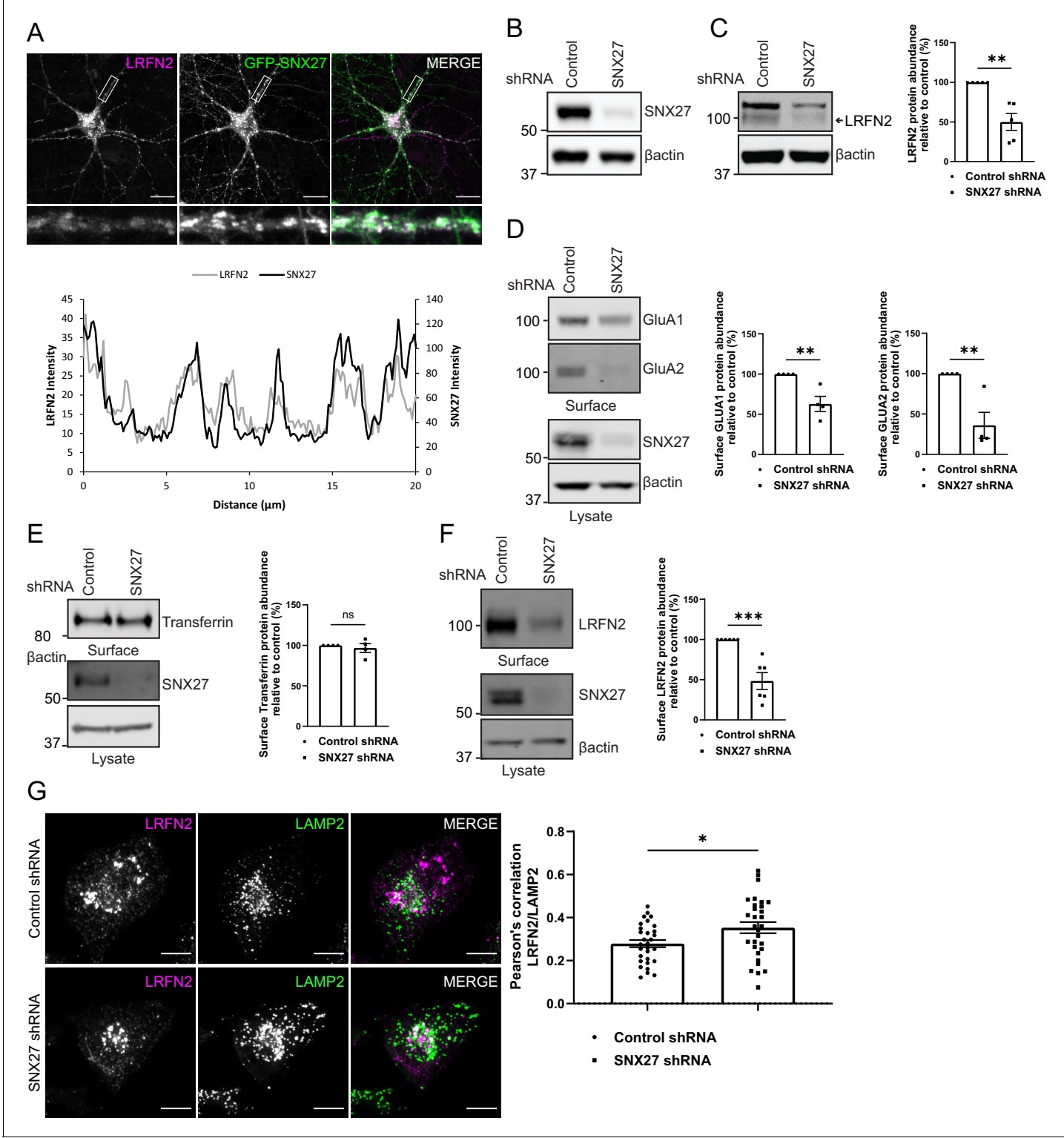

**Figure 3.** The membrane trafficking of LRFN2 is dependent on SNX27. (A) Immunofluorescence staining of DIV20 rat hippocampal neurons co-transduced with mCherry-LRFN2 (pseudocoloured in magenta) and GFP-SNX27 expressing sindbis viruses. Scale bars, 20 μm. White boxes indicate the 20 μm section zoomed in. Representative fluorescence intensity plot shown of the 20 μm zoomed in section from (i). (B–C) Fluorescence-based western analysis of DIV19 rat cortical neurons transduced with either a control or SNX27 shRNA for: (B) endogenous SNX27 (C) endogenous LRFN2. Actin was used as a protein load control. Quantification from five independent experiments (n = 5). Data expressed as a percentage of the control shRNA and analysed by an unpaired t-test. (D–E) Fluorescence-based western analysis after surface biotinylation and streptavidin agarose capture of membrane

*Figure 3 continued on next page*

*Figure 3 continued*

proteins of DIV19 rat cortical neurons transduced with either a control or SNX27 shRNA for: (D) Endogenous surface GluA1 and GluA2. Total levels of endogenous SNX27 are also shown. Quantification from four independent experiments (n = 4). Data expressed as a percentage of the control shRNA and analysed by an unpaired t-test. (E) Endogenous surface Transferrin receptor. Total levels of endogenous SNX27 are also shown. Quantification from four independent experiments (n = 4). Data expressed as a percentage of the control shRNA and analysed by an unpaired t-test. (F) Endogenous surface LRFN2. Total levels of endogenous SNX27 are also shown. Quantification from six independent experiments (n = 6). Data expressed as a percentage of the control shRNA and analysed by an unpaired t-test. (G) Immunofluorescence staining of internalised LRFN2 in H4 cells transduced with either control or SNX27 shRNA. Cells were transfected with mCherry-LRFN2 and after 24 hr the surface LRFN2 labelled using an mCherry antibody. The labelled mCherry-LRFN2 was allowed to internalise for 2 hr before fixation and permeabilisation. Cells were co-stained with LAMP2 as a lysosome marker. Scale bars, 10 μm. Quantification of colocalisation of LRFN2 (pseudocoloured in magenta) and LAMP2 (pseudocoloured in green) from three independent experiments (n = 30 cells analysed). Data analysed by an unpaired t-test. In all figures error bars represent mean ± SEM. ****, P≤0.001; **, P≤0.01; *, P≤0.05.

The online version of this article includes the following source data for figure 3:

**Source data 1.** Original immunoblots.
**Source data 2.** Data for *Figure 3C* – G.

examined the relative cell surface localisation of LRFN2 and AMPA receptors in neurons. Here, we transduced primary rat hippocampal neuronal cultures with sindbis virus encoding mCherry-tagged LRFN2 such that the mCherry tag was exofacially expressed. After 24 hr of expression, we observed the surface localisation of LRFN2 and AMPA receptors by indirect immunofluorescence using an mCherry antibody (and Alexa Fluor 647 labelled secondary antibody), and antibodies against extracellular epitopes of the endogenous AMPA receptor subunits GluA1 and GluA2, followed by fixation. This revealed points of overlap between the distribution of cell surface LRFN2 and endogenous GluA1 and GluA2 in dendrites (*Figure 4A and B and C*).

To investigate the possible association of LRFN proteins with AMPA receptors, we transiently co-transfected HEK293T cells with full-length mCherry-tagged versions of LRFN1, LRFN2, or LRFN4 and either full-length super-ecliptic pHluorin (SEP)-tagged GluA1 or GluA2. Twenty-four hr later GFP-nanotrap immunoisolation (targeting the SEP tag) and western analysis revealed that LRFN1, LRFN2, and LRFN4 were all able to associate with GluA1 and GluA2 (*Figure 4D*). To map the region responsible for this interaction, we engineered a series of amino-terminal truncations of the extracellular region of LRFN2 (*Figure 4E*). The resulting mCherry-tagged deletion mutants were transiently co-transfected into HEK293T cells alongside either SEP-GluA1 or myc-GluA2 encoding vectors prior to RFP-nanotrap immunoisolation (targeting the mCherry tag) and western analysis of associating proteins. While full-length LRFN2 associated with GluA1 and GluA2, deletion of the LRR and Ig domains led to a reduction in binding (an approximate 56% reduction for GluA1, n = 4, unpaired t-test, t (6) = 3.3, p = 0.0173; and an approximate 67% reduction for GluA2, n = 4, unpaired t-test, t (6) = 10.6, p = <0.0001) that was further reduced upon removal of the juxtamembrane fibronectin type III domain (an approximate 79% reduction for GluA1, n = 4, unpaired t-test, t (6) = 12.8, p = <0.0001; and an approximate 94% reduction for GluA2, n = 4, unpaired t-test, t (6) = 40.5, p = <0.0001) (*Figure 4F*). Taken together, this data demonstrates that LRFN2 and AMPA receptors show overlapping distributions on the cell surface of dendrites, and that LRFN2 associates with the GluA1 and GluA2 subunits of AMPA receptors through an interaction principally mediated by its extracellular LRR and Ig domains.

To investigate whether LRFN2 suppression affects the surface levels of GluA1 and GluA2, we turned to an immunofluorescence analysis in primary neurons. To relate this single-cell analysis with the previously described population-based biochemical quantification of cell surface AMPA receptors (see *Figure 3E*), we suppressed SNX27 expression through transduction of SNX27 targeting shRNA into primary rat cortical neuronal cultures. After 7 days of culturing, the intensity of GluA1 and GluA2 staining were quantified with antibodies that specifically detect extracellular epitopes of these subunits. Consistent with the aforementioned biochemical analysis this revealed a significant reduction in cell surface GluA1 and GluA2 staining upon SNX27 suppression (GluA1: an approximate 16% reduction, n = 50, Mann Whitney test, U (730), p = 0.0003; GluA2: an approximate 34% reduction n = 50, Mann Whitney test, U (632), p < 0.0001) (*Figure 5A*). In a parallel analysis, the suppression of LRFN2 expression (*Figure 5B*) also caused a significant reduction in GluA2 surface

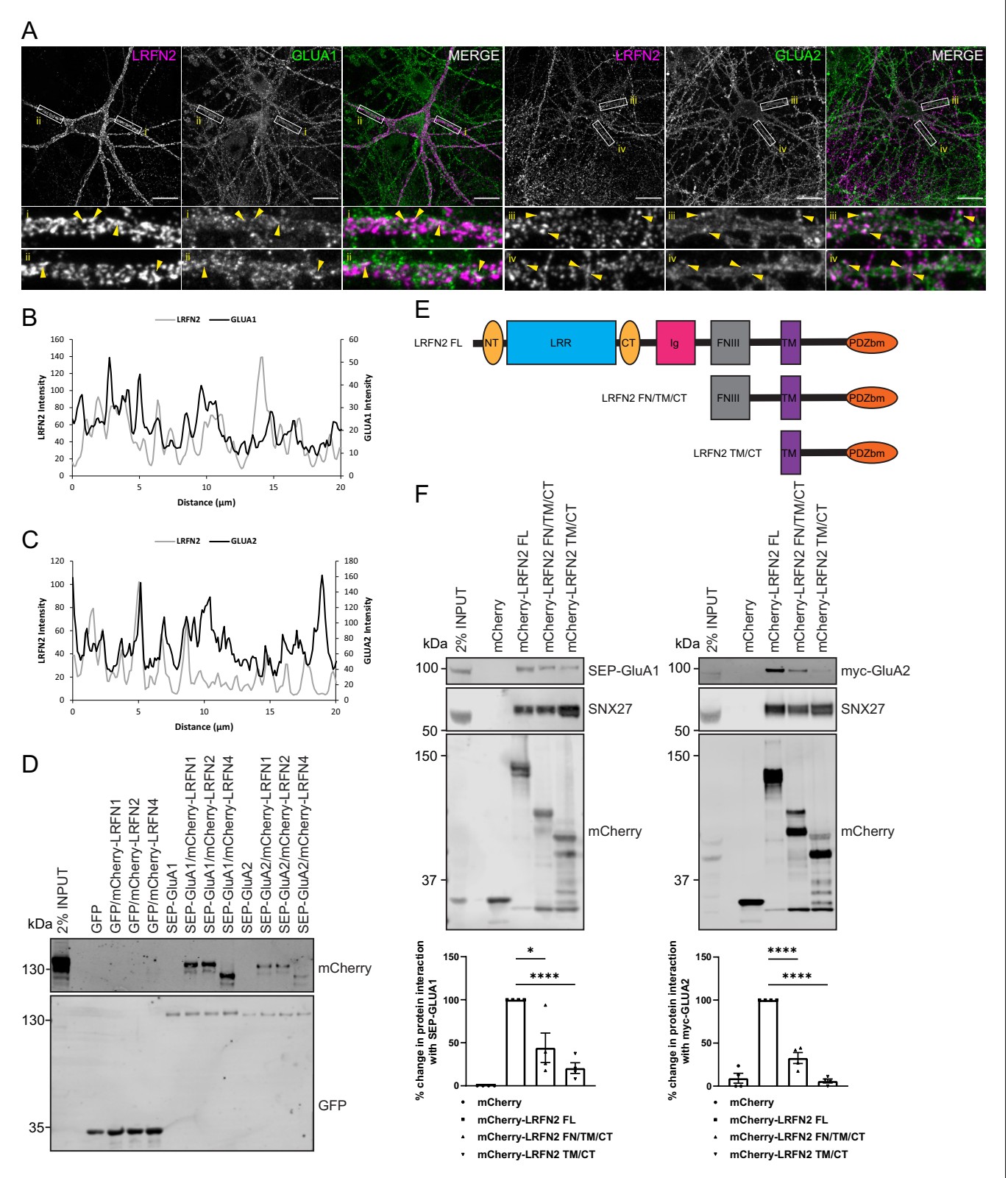

**Figure 4.** LRFNs interact with AMPA receptors. (**A**) Immunofluorescence staining of endogenous surface GluA1 and GluA2 in DIV20 rat hippocampal neurons transduced with mCherry-LRFN2 expressing sindbis virus. An mCherry antibody was used to stain for surface LRFN2 expression followed by a far-red labelled secondary antibody (pseudocoloured in magenta). Cells were co-stained with antibodies against extracellular epitopes of the endogenous AMPA receptor subunits GluA1 and GluA2 (shown in green) Scale bars, 20 μm. White boxes indicate the 20 μm section zoomed in. Yellow

*Figure 4 continued on next page*

*Figure 4 continued*
arrows show points of overlap. (**B**) Representative fluorescence intensity plots of the 20 µm zoomed in sections for surface LRFN2 and GluA1 in A (region i). (**C**) Representative fluorescence intensity plots shown of the 20 µm zoomed in sections for surface LRFN2 and GluA2 in (A; region iii). (**D**) Fluorescence-based western analysis after GFP-Trap immunoprecipitation of full-length SEP-GluA1 or SEP-GluA2 co-expressed with full-length mCherry-LRFN1, mCherry-LRFN2 or mCherry-LRFN4 in HEK293T cells. (**E**) Schematic of LRFN2 constructs mCherry-LRFN2 FL (full-length) and N-terminal mutants. LRR, leucine-rich repeat; NT/CT, N/C-terminal domains of LRR; Ig, immunoglobulin domain; FNIII, fibronectin type-III; TM, transmembrane; PDZbm, PDZ binding motif. (**F**) Fluorescence-based western analysis after RFP-Trap immunoprecipitation of mCherry-LRFN2 wild-type (LRFN2 FL) or N-terminal mutants co-expressed with full-length SEP-GluA1 or myc-GluA2 in HEK293T cells. Quantification from four independent experiments (n = 4). Data expressed as a percentage of the full-length mCherry-LRFN2 and analysed by an unpaired t-test. In all figures error bars represent mean ± SEM. ****, P≤0.0001; *, P≤0.05.

The online version of this article includes the following source data for figure 4:

**Source data 1.** Original immunoblots.
**Source data 2.** Data for *Figure 4F*.

expression (an approximate 40% reduction, n = 50, Mann Whitney test, U (556), p < 0.0001) but, interestingly, had no significant effect on the cell surface expression of GluA1 (n = 50, Mann Whitney test, U (1228), p = 0.8813) (*Figure 5C*).

To assess further the trafficking of the AMPA receptor subunit GluA2, we analysed the internalisation of GluA2 after suppression of SNX27 or LRFN2 in primary neurons. After 7 days of culturing with shRNA treatment, surface GluA2 receptor subunits were labelled with an antibody that specifically detected the extracellular epitope of the GluA2 subunit. The GluA2 was allowed to internalise for 1 hr before fixation. The antibody remaining on the surface was labelled with a secondary Alexa Fluor 405 antibody. After permeabilisation, the neurons were co-stained with VPS35 to label the endosome or LAMP1 to label the lysosome and internalised receptors were detected using a distinct secondary Alexa Fluor 647 antibody. The intensity of internalised GluA2 was quantified in the cell body and proximal dendrites and found to be significantly increased after suppression of both SNX27 (an approximate 43% increase, n = 58, Mann Whitney test, U (827), p < 0.0001) and LRFN2 (an approximate 14% increase, n = 58, Mann Whitney test, U (1198), p = 0.0073) with the GluA2 overlapping with both VPS35 and LAMP1 (*Figure 6A, B, C*). This suggests that suppression of SNX27 or LRFN2 perturbs the recycling of GluA2, increasing its retention time in the endo-lysosomal system. Overall, these data establish a biochemical and functional connection between LRFN2 and the SNX27-dependent membrane trafficking of the AMPA receptor subunit GluA2.

## In vivo suppression of SNX27 and LRFN2 affects ex vivo AMPA receptor activity in the hippocampus

In isolated neuronal cultures SNX27 and LRFN2 suppression regulates AMPA receptor membrane trafficking and cell surface expression. To relate these phenotypes to the functional activity of synaptic AMPA receptors, we turned to an electrophysiological analysis in ex vivo slices. Here, we stereotaxically injected one dorsal hippocampal hemisphere of an adult rat with lentivirus encoding for specific targeting shRNAs (targeting either SNX27 or LRFN2) and injected the other hemisphere of the same animal with a control non-targeting shRNA lentivirus (all lentiviruses were engineered to express GFP in order to visualise the transduced area). Six to eight weeks after surgery, we prepared ex vivo hippocampal slices and performed electrophysiological recordings by stimulating Schaffer collaterals and recording field excitatory postsynaptic potentials (fEPSPs) in the stratum radiatum of GFP-positive regions of CA1 (*Figure 7A*). Input-output curves showed that SNX27 shRNA treatment profoundly decreased excitatory synaptic transmission compared to the non-targeting control shRNA (two-way repeated-measures ANOVA, between subjects, F (1,19) = 22.3, p = 0.0001) (*Figure 7B and C*). Importantly, LRFN2 shRNA treatment also significantly decreased excitatory synaptic transmission compared to the non-targeting control shRNA (two-way repeated-measures ANOVA, between subjects F (1,12) = 6.7, p = 0.024) (*Figure 7D and E*). A direct comparison of LRFN2 and SNX27 phenotypes revealed that SNX27 produced a larger reduction in glutamatergic transmission than LRFN2 suppression (two-way repeated-measures ANOVA, between subjects F (1,16) = 4.6, p = 0.047) (*Figure 7F*), whereas there was no difference in transmission between the two non-targeting control shRNAs (F (1,15) = 0.95, p = 0.35), suggesting that SNX27, as well as

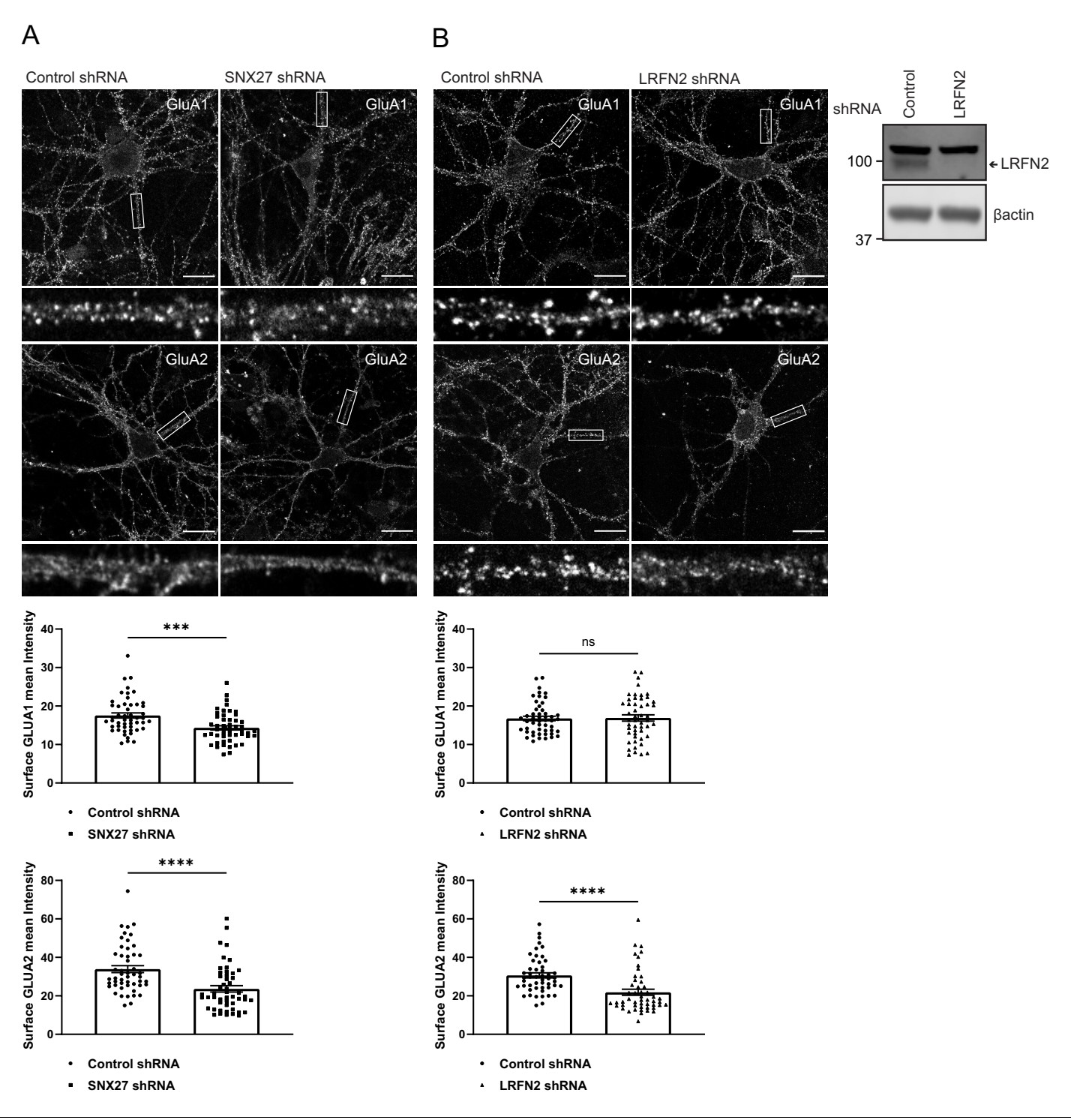

**Figure 5.** SNX27 and LRFN2 suppression affects surface AMPA receptor expression. (**A**) Immunofluorescence staining of endogenous surface GluA1 and GluA2 in DIV19 rat hippocampal neurons transduced with either control or SNX27 shRNA. Scale bars, 20 μm. White boxes indicate the 20 μm section zoomed in. Quantification of surface GluA1 and GluA2 from five independent experiments (n = 50 neurons analysed). Data analysed by a Mann-Whitney U test. (**B**) Immunofluorescence staining of endogenous surface GluA1 and GluA2 in DIV19 rat hippocampal neurons transduced with either control or LRFN2 shRNA. Scale bars, 20 μm. White boxes indicate the 20 μm section zoomed in. Quantification of surface GluA1 and GluA2 from five independent experiments (n = 50 neurons analysed). Data analysed by a Mann-Whitney U test. In all figures error bars represent mean ± SEM. ****, p≤0.0001; ***, p≤0.001; ns, not significant. Fluorescence-based western analysis shown of DIV19 rat cortical neurons transduced with either control or LRFN2 shRNA for endogenous LRFN2. Actin was used as a protein load control.

*Figure 5 continued on next page*

*Figure 5 continued*

The online version of this article includes the following source data for figure 5:

**Source data 1.** Original immunoblots.
**Source data 2.** Data for *Figure 5A and B*.

affecting LRFN2 expression, may also be regulating other LRFNs and/or other synaptic proteins that can influence AMPA receptor activity. We are however cautious in over-interpreting such a comparison.

Finally, loss of SNX27 has previously been shown to impair induction of hippocampal LTP (*Wang et al., 2013*). Therefore, we asked whether LRFN2 suppression also affects activity-dependent synaptic plasticity. Following high-frequency stimulation (HFS), we observed that the non-targeting control-treated slices exhibited robust LTP (paired t-test, t (5) = 3.9, p = 0.011) (*Figure 7G and H*) while no detectable LTP was induced in the LRFN2 shRNA-treated slices (paired t-test, t (4) = 1.8, p = 0.15) (*Figure 7G and I*) indicating that LRFN2 suppression attenuates the induction of LTP. Taken together these data indicate that LRFN2 plays an important role in the functional expression and activity of synaptic AMPA receptors and reveals a new player in the complex SNX27-mediated regulation of AMPA receptor trafficking.

## Discussion

Patients lacking SNX27 expression or expressing predicted damaging inherited SNX27 variants display a range of neuronal phenotypes that include developmental delays, abnormal neurocognitive function, epilepsy, various types of seizure, and subcortical white matter abnormalities (*Damseh et al., 2015*; *Parente et al., 2020*). While known SNX27-associated neuronal integral proteins, such as NMDA receptors (*Cai et al., 2011*; *Wang et al., 2006*), 5-HT4 receptor (*Joubert et al., 2004*), metabotropic glutamate receptor 5 (mGluR5) (*Lin et al., 2015*), neuroligin 2 (*Binda et al., 2019*; *Halff et al., 2019*), and Kir3 channels (*Lunn et al., 2007*), have provided some insight into these complex phenotypes our unbiased quantitative identification of the neuronal SNX27 interactome has revealed an additional cohort of integral neuronal proteins that associate with SNX27. Besides LRFN2 (see discussion below), many of these proteins function in an array of neuronal activities: the high affinity glutamate transporter, SLC1A3, and the sodium bicarbonate co-transporter, SLC4A7, are associated with controlling glutamate neurotoxicity *Rothstein et al., 1996*; *Watase et al., 1998*; *Jen et al., 2005*; *Park et al., 2019*; SLC6A11, a sodium-dependent transporter, uptakes GABA and modulates GABAergic tone *Borden, 1996*; KCNT2, an outward rectifying potassium channel, is associated with early infantile epileptic encephalopathies *Bhattacharjee et al., 2003*; *Gururaj et al., 2017*; KIDINS220, a scaffold in neurotrophin signalling, is associated with spastic paraplegia and intellectual disability *Kong et al., 2001*; *DDD Study et al., 2016*; and ADAM22 is a receptor for the neuronal secreted protein LGI1 (*Fukata, 2006*), the product of the causative gene for autosomal dominant partial epilepsy with auditory features (*Muona et al., 2016*). This significant expansion in the neuronal targets for SNX27-mediated endosomal sorting has therefore broadened our understanding of those integral proteins whose perturbed cell surface expression may underly the complex neurological phenotypes observed in SNX27-associated pathologies.

A major functional role for neuronal SNX27 is as an established regulator of postsynaptic AMPA receptor trafficking (*Wang et al., 2013*; *Loo et al., 2014*; *Hussain et al., 2014*). AMPA receptors are one of the major types of ionotropic glutamate receptors present at the postsynaptic density of excitatory synapses and, consequently, perturbed AMPA receptor trafficking has been implicated in numerous neurological and psychiatric disorders (*Henley and Wilkinson, 2016*, *Shankar et al., 2008*, *Shepherd and Huganir, 2007*, *Walsh et al., 2002*, *Grooms et al., 2000*, *Yamashita and Kwak, 2014*). Four core AMPA receptor subunits, GluA1 through to GluA4, combine to produce functionally diverse homo- and hetero-tetrameric channels which drive fast excitatory synaptic transmission (*Hollmann and Heinemann, 1994*, *Henley and Wilkinson, 2016*). AMPA receptors utilise membrane trafficking pathways to dynamically regulate their number, composition, and biophysical properties at the postsynaptic membrane during the modulation of synaptic strength associated with learning and memory (*Henley and Wilkinson, 2016*; *Diering and Huganir, 2018*; *Huganir and*

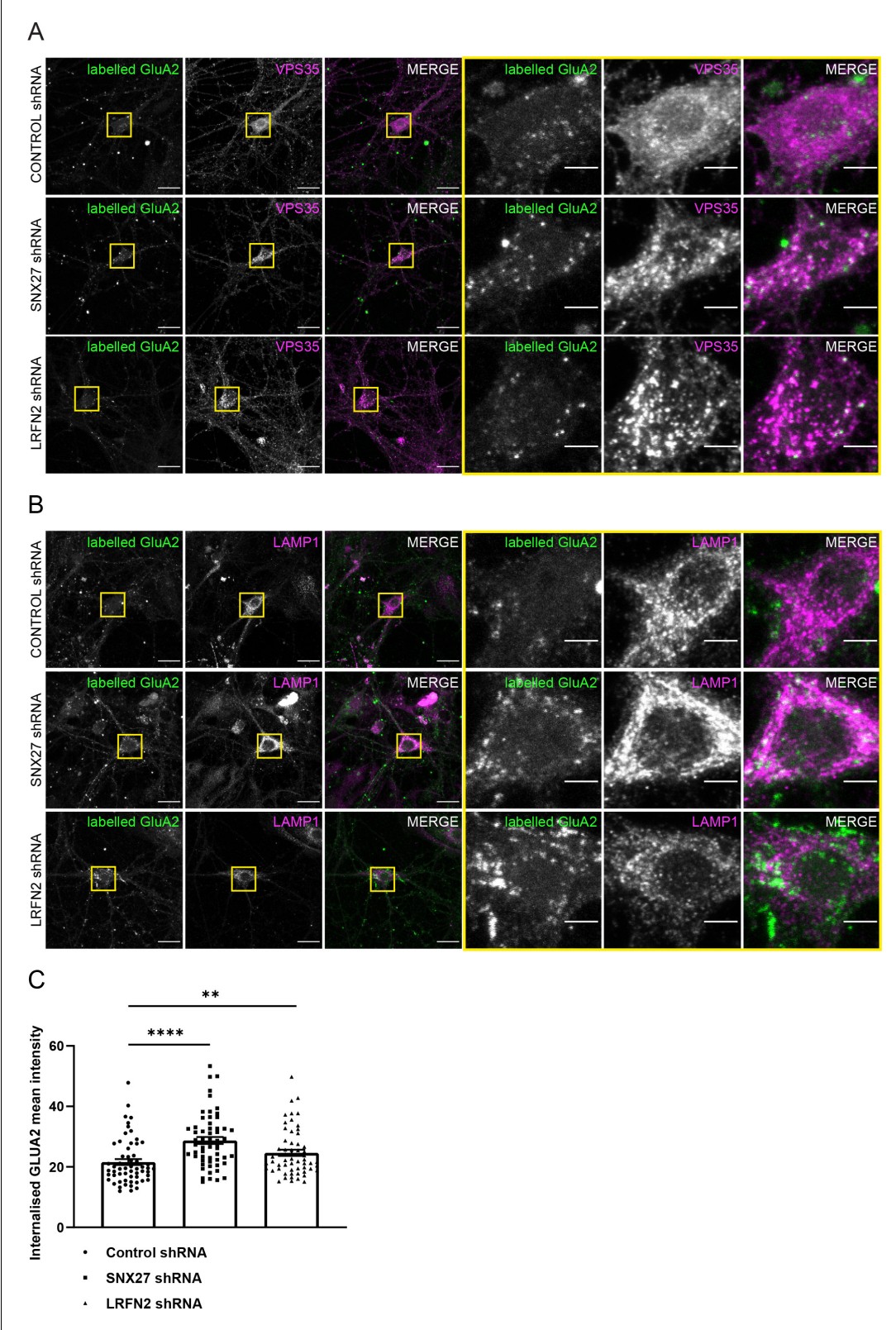

**Figure 6.** Suppression of SNX27 or LRFN2 increases AMPA receptors in the endo-lysosomal system. (**A–B**) Immunofluorescence staining of endogenous internalised GluA2 (pseudocoloured in green) in DIV19 rat hippocampal neurons transduced with either control, SNX27 or LRFN2 shRNA. Surface labelled GluA2 was allowed to internalise for 1 hr before fixation and permeabilisation. Scale bars, 20 μm. Zoomed in images of the cell body (Yellow boxes). Scale bars, 5 μm. For clarity only the internalised GluA2 images are shown. The surface GluA2 expression can be found in *Figure 6—figure*

*Figure 6 continued on next page*

*Figure 6 continued*

*supplement 1*. (A) Cells were co-stained with VPS35 (pseudocoloured in magenta) as an endosomal marker. (B) Cells were co-stained with LAMP1 (pseudocoloured in magenta) as a lysosome marker. (C) Quantification of internalised GluA2 intensity from five independent experiments (n = 58 neurons analysed). The mean intensity of GluA2 was measured from the cell body and proximal dendrites. Data analysed by a Mann-Whitney U test. In all figures error bars represent mean ± SEM. ****, P≤0.0001; **, P≤0.01.

The online version of this article includes the following source data and figure supplement(s) for figure 6:

**Source data 1.** Data for *Figure 6C*.

**Figure supplement 1.** Suppression of SNX27 or LRFN2 increases AMPA receptors in the endo-lysosomal system.

*Nicoll, 2013*). At the mechanistic level, the regulation of AMPA receptors by SNX27 is considered to arise from the PDZ domain of SNX27 directly binding to the PDZ binding motifs present in the carboxy-terminal tails of AMPA receptor subunits. However, using unbiased proteomics and biochemical validation coupled with in cellulo and ex vivo analysis, we have presented evidence consistent with SNX27 regulating the surface expression and activity of AMPA receptors through association with the synaptic adhesion protein LRFN2 (*Figure 8*). LRFN2 belongs to the LRFN family, also known as synaptic adhesion-like molecules (SALMs), and whilst these proteins have been associated with inhibitory synapses (*Li et al., 2018*) and at the pre-synapse (*Brouwer et al., 2019*), their function has principally been linked with excitatory synapses (*Ko et al., 2006*; *Morimura et al., 2006*; *Wang et al., 2006*; *Lie et al., 2018*; *Mah et al., 2010*; *Loh et al., 2016*). In proposing LRFN2 as an accessory protein in the dynamic AMPA receptor trafficking code (*Diering and Huganir, 2018*), our data adds to the complex regulation of AMPA receptor-mediated synaptic transmission and plasticity.

We have established that LRFN2 directly interacts with SNX27 and that this is required for its trafficking to the plasma membrane and away from degradation in the lysosome. In addition, we have shown that LRFN2 associates with the AMPA receptor subunits GluA1 and GluA2, interactions that are principally governed by the extracellular LRR and Ig domains of LRFN2. This is consistent with these domains mediating protein:protein interactions, most notably for the trans-synaptic interactions of LRFNs with type-II receptor tyrosine phosphatases (*Lin et al., 2018*). In the hippocampus the major forms of AMPA receptors include GluA1/2 and GluA2/3 heteromers, and GluA1 homomers, with the differential trafficking of GluA1-containing and GluA1-lacking receptors having important implications in transmission and plasticity (*Diering and Huganir, 2018*; *Huganir and Nicoll, 2013*; *Wenthold et al., 1996*). Our data indicates that LRFN2 helps maintain surface expression of GluA1-lacking receptors, but suggests that other pathways that also rely on SNX27 may exist to maintain surface expression of GluA1-containing receptors. Our focus in this study was LRFN2 as it was the only LRFN retained through our high stringency data filtering, and because defects in LRFN2 have been implicated in various neurological conditions, consistent with an important role in brain development, function, and cognition (*Morimura et al., 2017*; *Rautiainen et al., 2016*; *Thevenon et al., 2016*). However, given that LRFN1 and LFRN4 also associated with SNX27 and contain an optimal PDZ binding motif for binding to the SNX27 PDZ domain, we consider that the endosomal sorting of these proteins will also by mediated by SNX27 and that further studies into LRFN1 and LRFN4 are therefore likely to provide additional insights into SNX27-associated pathologies. Interestingly, LRFN1 has also been shown to interact with AMPA receptors on the synaptic surface (*Morimura et al., 2017*). It is therefore tempting to speculate that redundancies between the LRFN proteins may explain why GluA1-containing receptors are not affected after LRFN2 suppression.

Using electrophysiology in ex vivo slices, we have shown that suppression of SNX27 leads to a dramatic loss of AMPA receptor-mediated synaptic activity, validating our biochemical data showing reduced surface AMPA receptor expression. This supports previous findings showing reduced AMPA receptor-mediated postsynaptic currents in SNX27$^{+/-}$ mice (*Wang et al., 2013*). Importantly, we have also shown that LRFN2 depletion caused a loss of AMPA receptor-mediated synaptic activity, consistent with a role for LRFN2 in SNX27-dependent AMPA receptor trafficking. We also observed an attenuated activity-dependent hippocampal LTP upon LRFN2 suppression, which supports the study of Li and colleagues who reported a reduction of LTP in an LRFN2 knockout mouse model (*Li et al., 2018*). In contrast, Morimura and colleagues reported an increase in silent synapses

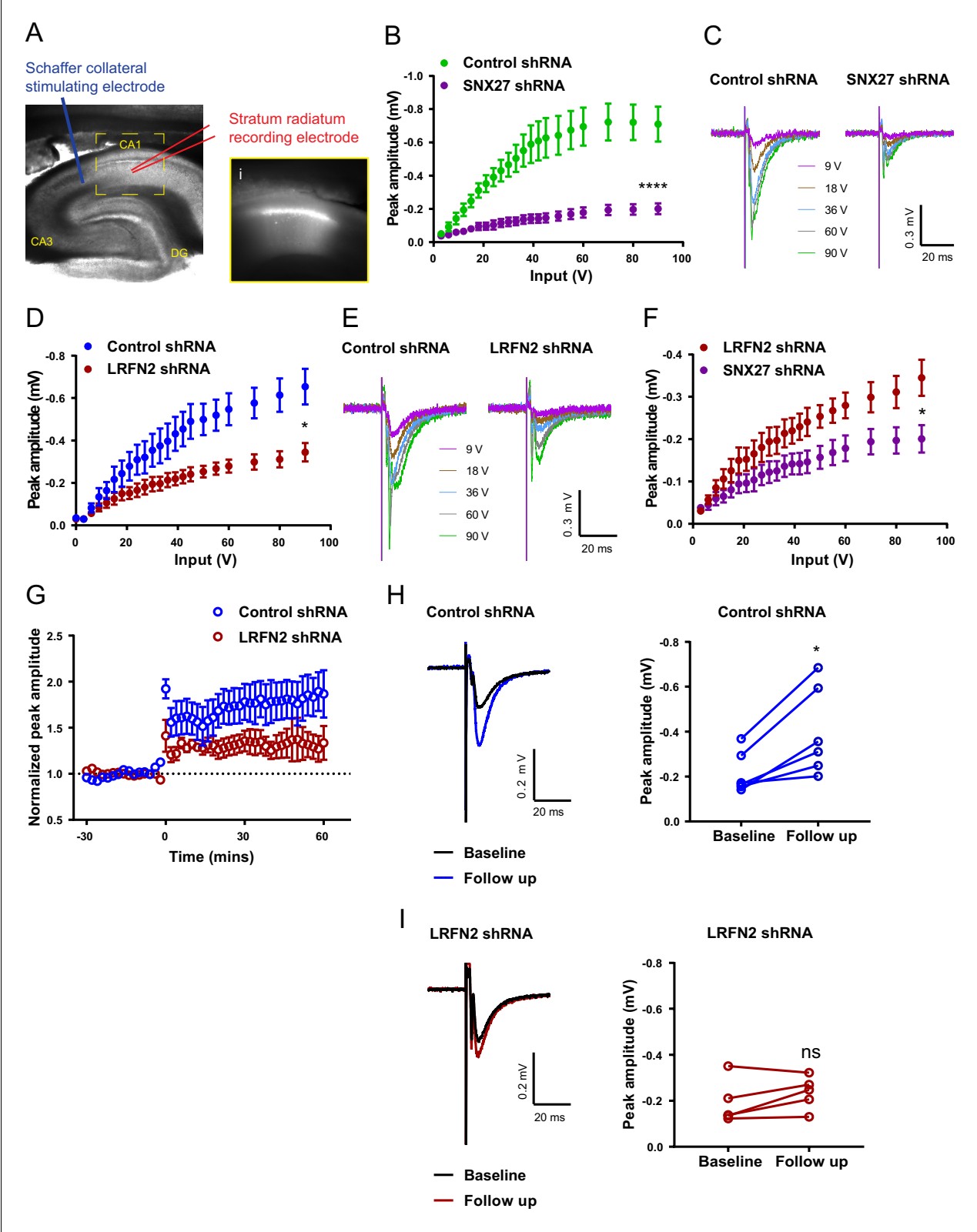

**Figure 7.** In vivo suppression of SNX27 and LRFN2 affects ex vivo AMPA receptor activity in the hippocampus. (**A**) Representative infrared image of a transduced rat dorsal hippocampal slice expressing the control shRNA lentivirus showing placement of electrodes to record Schaffer collateral responses. Inset (i) shows GFP fluorescence of CA1 region. Animals were left for 6–8 weeks with the control shRNA injected into one hemisphere and either SNX27 or LRFN2 shRNA into the other hemisphere. (**B**) Input-output curves from SNX27 and control shRNA treated slices. Quantification from six

*Figure 7 continued on next page*

*Figure 7 continued*

animals (n = 10 recordings) for the control and five animals (n = 11 recordings) for the SNX27 shRNA. Data analysed by a two-way repeated-measures ANOVA. (C) Representative traces of the recordings from (B). (D) Input-output curves from LRFN2 and control shRNA-treated slices. Quantification from five animals (n = seven recordings) for the control and five animals (n = seven recordings) for the LRFN2 shRNA. Data analysed by a two-way repeated-measures ANOVA. (E) Representative traces of the recordings from (D). (F) Input-output curves comparing SNX27 and LRFN2 shRNA treatment. (G) LTP response following high frequency stimulation (HFS) at t = 0. Quantification from five animals (n = six recordings) for the control and four animals (n = five recordings) for the LRFN2 shRNA. (H) Representative traces and quantification of the LTP response after treatment with the control shRNA (follow-up is the final 10 min of the recording). Control slices underwent LTP with data analysed by a paired t-test. (I) Representative traces and quantification of the LTP response after treatment with the LRFN2 shRNA. LRFN2 shRNA treated slices were not significantly potentiated. In all figures error bars represent mean ± SEM. ****, p≤0.0001; *, p≤0.05; ns, not significant.

The online version of this article includes the following source data for figure 7:

**Source data 1.** Data for *Figure 7B* - I.

and an enhancement of LTP in another LRFN2 knockout mouse model (*Morimura et al., 2017*). The distinct methodology used in our study, post-development suppression of LRFN2 compared with developmental knockout of LRFN2, may in part explain these differences.

It is well established that AMPA receptors can interact with a multitude of transmembrane proteins to stabilise and retain their expression at the postsynaptic density, including scaffolding proteins, synaptic adhesion molecules, and auxiliary subunits, such as the transmembrane AMPA receptor regulatory proteins (TARPs) and cornichon-like proteins CNIH2/CNIH3 (*Schwenk et al., 2009*; *Coombs and Cull-Candy, 2009*). Whilst we have shown that LRFN2, analogous to these auxiliary proteins, plays important roles in the targeting of AMPA receptors to the post-synapse, we have not examined the question of whether LRFN2 clusters AMPA receptors into slots at the postsynaptic density (*Henley and Wilkinson, 2016*) nor have we investigated the trafficking of LRFN2 through the biosynthetic pathway and how this may regulate LRFN2 surface levels (*Seabold et al., 2012*). The PDZ binding motif present in LRFN1, LRFN2 and LRFN4 can associate with many other PDZ domain-containing proteins including the synaptic scaffolding protein PSD-95 (*Ko et al., 2006*; *Morimura et al., 2006*; *Nam et al., 2011*; *Wang et al., 2006*), suggesting that LRFNs could also have a role in the anchoring and stability of AMPA receptors at the synaptic surface. It is possible that LRFNs interact with the AMPA receptor subunits through their extracellular LRR and Ig domains and PSD-95 through their PDZ binding motifs to stabilise AMPA receptors at the synaptic surface (*Figure 8*). This would be an interesting avenue to explore further.

Interestingly, NMDA receptors, which are also required for excitatory transmission and synaptic plasticity, interact with both SNX27 and LRFN2 (*Wang et al., 2006*; *Thevenon et al., 2016*; *Morimura et al., 2006*; *Cai et al., 2011*; *Wang et al., 2013*; *Clairfeuille et al., 2016*). The association of SNX27 with LRFN2 may therefore play multiple roles in regulating excitatory transmission through regulating surface expression of glutamate receptors at the synaptic surface.

LRFN2 is increasingly being linked to a range of neurological conditions including antisocial personality disorder (*Rautiainen et al., 2016*), autism (*Morimura et al., 2017*), schizophrenia (*Morimura et al., 2017*), working memory deficits, and learning disabilities (*Thevenon et al., 2016*). A recent proteomic screen on pre-frontal *post-mortem* tissue also revealed that LRFN2 protein expression was significantly decreased in people with Alzheimer's disease, Parkinson's disease with dementia, and dementia with Lewy bodies, further highlighting the importance of LRFN2 for neuronal function (*Bereczki et al., 2018*). LRFN2 had the highest level of reduction compared to other synaptic proteins in all three forms of dementia, and its loss was strongly associated with rate of cognitive decline (*Bereczki et al., 2018*). Our data demonstrating that LRFN2 can control AMPA receptor activity therefore provides additional insight into the role of LRFN2 in these conditions.

In summary, we have identified LRFN2 as a high-affinity neuronal interactor of SNX27 that is required for AMPA receptor-mediated synaptic transmission. Suppression of SNX27 leads to a reduction in LRFN2 expression which results in a loss of surface expression of GluA2 and, correspondingly, a loss of synaptic activity (*Figure 8*). The link between SNX27, LRFN2, and AMPA receptors provides a new point of control for AMPA receptor-mediated synaptic transmission and plasticity and provides additional insight into the association of LRFN2 and SNX27 with many neurological and psychiatric conditions.

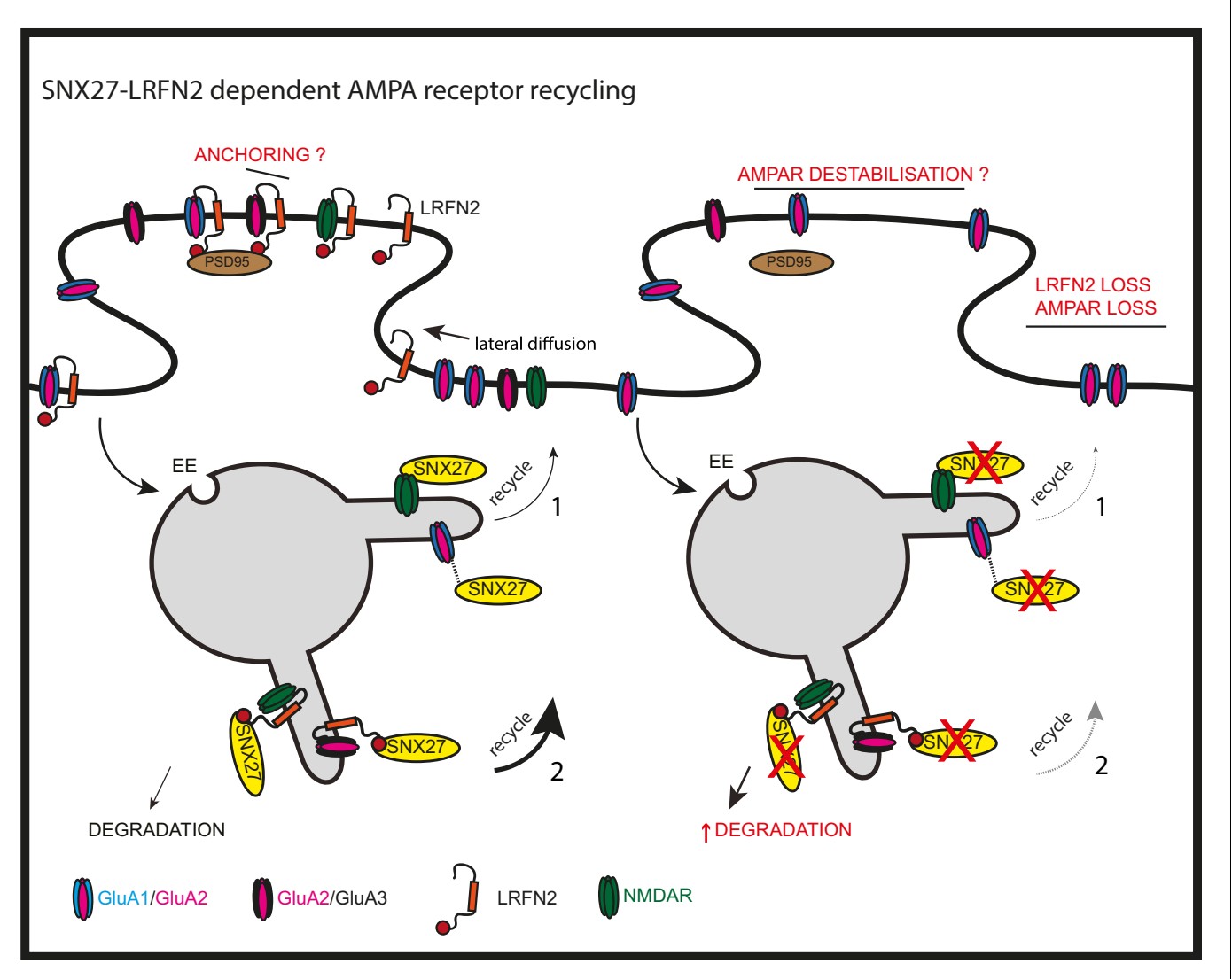

**Figure 8.** SNX27 regulation of AMPA receptor trafficking through LRFN2 It is well established that SNX27 suppression results in a loss of AMPA receptor expression and activity through a mechanism that evokes the direct sequence-dependent recognition of internalised AMPA receptors by endosome associated SNX27 (1). In the absence of SNX27, internalised AMPA receptors fail to be recycled and enter the lysosomal degradative compartment. This missorting leads to an overall reduction in AMPA receptor expression at the cell surface. Our data proposes that LRFN2 acts as a 'bridging' factor in SNX27-mediated AMPA receptor trafficking (2). Here, internalised LRFN2 directly associates with SNX27 through its PDZ binding motif (PDZbm) and is a classical cargo for SNX27-dependent retrieval and recycling to the synaptic cell surface. Upon SNX27 suppression, internalised LRFN2 is missorted to lysosomes where it undergoes degradation, leading to a reduction in LRFN2 levels at the synaptic surface. LRFN2 associates with AMPA receptors and LRFN2 suppression leads to AMPA receptor missorting into the degradative pathway. Taken together, we propose that the association of LRFN2 with AMPA receptors is required for the SNX27-dependent endosomal recycling of GluA1-lacking AMPA receptors. LRFN2 may also display properties of an 'auxiliary' protein in aiding the retrieval and recycling of internalised AMPA receptors through the SNX27-positive endosomal compartment and, in addition, may function as an adhesion molecule to anchor and stabilise AMPA receptor expression at the cell surface. EE, early endosome; LE, late endosome.

## Materials and methods

### Plasmids

The intracellular C-terminal tails of GluA1 (amino acids 827–907), GluA2 (amino acids 834–883), GluA3 (amino acids 839–888), and GluA4 (amino acids 835–902) were cloned from rat whole brain cDNA into the vector pEGFP-C3. The intracellular C-terminal tails of wild type human LRFN1, LRFN2, LRFN3, SLC1A3, and SLC4A7 (last 40 amino acids) were cloned from HeLa cDNA into the

vector pEGFP-C1. The human LRFN4 (last 40 amino acids) was cloned from LRFN4 cDNA (MHS6278-202832013, GE Healthcare) and the human LRFN5 (last 40 amino acids) generated by combining oligonucleotides ordered from Eurofins genomics. The ΔPDZbm mutants (missing the last three amino acids) were cloned either directly from cDNA or by site-directed mutagenesis of the wild type plasmids. The LRFN2(pE786A) and LRFN2(pV789A) mutants were generated using site-directed mutagenesis of the LRFN2 wild-type plasmid. The LRFN2 full length construct and N-terminal mutants were cloned from HeLa cDNA and inserted into a pmCherryC1 vector containing the LRFN1 signal peptide upstream of the mCherry tag. Flag-tagged rat SNX27 was produced by PCR amplification of the SNX27 coding sequence from cDNA produced from PC12 cells with a FLAG tag added to the forward primer. The resulting PCR product was then cloned into the vector pcDNA3.1. SEP-GluA1 (*Blanco-Suarez and Hanley, 2014*) (64942; Addgene), SEP-GluA2 (*Ashby et al., 2004*) (64941; Addgene), and N-terminally myc-tagged GluA2 (*Leuschner and Hoch, 1999*) constructs have been reported previously.

## Cell culture

All cells were cultured in a humidified incubator at 37°C and 5% $CO_2$. HEK293T (American Type Culture Association (ATCC)) and H4 cells (a gift from Dr Helen Scott and Professor James Uney) were maintained in DMEM (D5796; Sigma-Aldrich) supplemented with 10% foetal bovine serum (F7524; Sigma-Aldrich). For the GFP/RFP-based immunoprecipitations, HEK293T cells were transfected with GFP/RFP- expressing constructs using polyethyleneimine (PEI) (Sigma-Aldrich). BHK-21 cells (ATCC) were maintained in Alpha MEM (22561–021, Gibco) supplemented with 5% foetal bovine serum (F7524; Sigma-Aldrich) and 1% Penicillin/Streptomycin (P0781, Sigma-Aldrich).

Primary neuronal cultures were prepared from embryonic day E18 Wistar rat brains as previously described (*Martin and Henley, 2004*). In brief, dissociated cortical cells were grown in six well dishes (500,000 cells/well), and hippocampal cells on 22 mm glass coverslips (150,000 cells/coverslip) coated with poly-L-lysine (P2636; Sigma-Aldrich) in 2 ml plating medium (Neurobasal medium (21103–049, Gibco) supplemented with 5% horse serum (H1270), 2% B27 (17504–044, Gibco) and 1% Glutamax (35050–038)) which was exchanged for 2 ml feeding medium 2 hr after plating (Neurobasal medium (21103–049, Gibco), 2% B27 (17504–044, Gibco), and 0.4% Glutamax (35050–038)). Cells were then fed with an additional 1 ml of feeding medium 7 days after plating.

## Lentivirus and Sindbis virus production

For lentivirus production, shRNAs driven by a H1 promoter were generated for the knockdown of rat SNX27 (target sequence 5'-aagaacagcaccacagaccaa-3') (*Binda et al., 2019*), human SNX27 (target sequence 5'- aagaacagtactacagaccaa-3'), rat LRFN2 (target sequence 5'-acgacgaggtactgattta-3'), and a control (non-targeting sequence 5'-aattctccgaacgtgtcac-3'). Oligonucleotides were cloned into a modified pXLG3-GFP viral vector and co-transfected into a 15 cm dish of HEK293T cells with the helper plasmids Pax2/p8.91 and pMDG2 using PEI. For primary culture, the viruses were harvested 72 hr after transfection, spun down at 4000 rpm for 10 min at room temperature (RT) and filtered through 0.45 µm filters before being stored at −70°C. Neurons were transduced with shRNA viruses on DIV12 and left for 7 days before analysis. Only those experiments where knockdown resulted in more than 85% reduction of the protein of interest were used for analysis. For in vivo injections, the lentiviral constructs were modified to include a Woodchuck Hepatitis Virus Posttranscriptional Regulatory Element (WPRE) to increase the GFP expression and the amount of HEK293T cells were scaled up to 10 x 15 cm dishes/virus. The media was harvested 48 hr after transfection, spun down at 4000 rpm for 10 min at RT and filtered through 0.45 µm filters. The filtered supernatant was then centrifuged in JA20 tubes at 6000 rpm overnight (O/N) at 4°C (Avanti J-25, Beckman Coulter). The following day the viral pellet was resuspended in 5 ml PBS and centrifuged at 20,000 rpm for 90 min at 4°C (Optima XL-100K, Beckman Coulter). The pellet was then resuspended in the required volume of PBS before being aliquoted and stored at −70°C.

For sindbis virus production, full-length human SNX27 and LRFN2 were cloned into the pSinRep5 plasmid. GFP-SNX27 was amplified from pEGFP-C1 and the resulting PCR product cloned into pSinRep5. The LRFN2 was subcloned from a pmCherryC1 vector which contained a mCherry fluorescent tag immediately after a N -terminal signal peptide (LRFN1 signal peptide) followed by full-length LRFN2. GFP- and mCherry- expressing pSinRep5 plasmids were created as controls. Five µg of in

vitro-transcribed RNA (2.5 µg of SNX27/LRFN2 RNA and 2.5 µg of the defective helper plasmid) was electroporated into 0.6x10^7 BHK-21 cells using a Gene Pulser II electroporation system (BioRad) in a gene pulser cuvette (0.2 cm gap). The electroporation conditions were set as follows: voltage; 1.5 kV, capacitance; 25 µF for a period of 0.7–0.8 ms. The viruses were harvested 36–48 hr after electroporation before being stored at −70°C. Neurons were transduced with sindbis viruses on DIV19/20 and left for 18–24 hr before analysis.

## TMT proteomics

Immunoisolated samples were reduced (10 mM TCEP, 55°C for 1 hr), alkylated (18.75 mM iodoacetamide, RT for 30 min) and then digested from the beads with trypsin (2.5 µg trypsin: 37°C, O/N). The resulting peptides were then labelled with TMT six plex reagents according to the manufacturer's protocol (Thermo Scientific) and the labelled samples pooled and desalted using SepPak cartridges according to the manufacturer's instructions (Waters). Eluate from the SepPak cartridge was evaporated to dryness and resuspended in 1% formic acid prior to analysis by nano-LC MSMS using an Ultimate 3000 nano-LC system in line with an LTQ-Orbitrap Velos mass spectrometer (Thermo Scientific).

In brief, peptides in 1% (vol/vol) formic acid were injected onto an Acclaim PepMap C18 nano-trap column (Thermo Scientific). After washing with 0.5% (vol/vol) acetonitrile 0.1% (vol/vol) formic acid peptides were resolved on a 250 mm × 75 µm Acclaim PepMap C18 reverse phase analytical column (Thermo Scientific) over a 150 min organic gradient, using seven gradient segments (1–6% solvent B over 1 min., 6–15% B over 58 min., 15–32% B over 58 min., 32–40% B over 5 min., 40–90% B over 1 min., held at 90% B for 6 min and then reduced to 1% B over 1 min.) with a flow rate of 300 nl min$^{-1}$. Solvent A was 0.1% formic acid and Solvent B was aqueous 80% acetonitrile in 0.1% formic acid. Peptides were ionised by nano-electrospray ionisation at 2.0 kV using a stainless-steel emitter with an internal diameter of 30 µm (Thermo Scientific) and a capillary temperature of 250°C. Tandem mass spectra were acquired using an LTQ- Orbitrap Velos mass spectrometer controlled by Xcalibur 2.1 software (Thermo Scientific) and operated in data-dependent acquisition mode. The Orbitrap was set to analyse the survey scans at 60,000 resolution (at m/z 400) in the mass range m/z 300–1800 and the top 10 multiply charged ions in each duty cycle selected for MS/MS fragmentation using higher energy collisional dissociation (HCD) with normalised collision energy of 45%, activation time of 0.1 ms and at a resolution of 7500 within the Orbitrap. Charge state filtering, where unassigned precursor ions were not selected for fragmentation, and dynamic exclusion (repeat count, 1; repeat duration, 30 s; exclusion list size, 500) were used.

The raw data files were processed and quantified using Proteome Discoverer software v2.1 (Thermo Scientific) and searched against the UniProt Rat database (downloaded January 2019: 35759 entries) using the SEQUEST algorithm. Peptide precursor mass tolerance was set at 10 ppm, and MS/MS tolerance was set at 0.6 Da. Search criteria included oxidation of methionine (+15.995 Da), acetylation of the protein N-terminus (+42.011 Da) and Methionine loss plus acetylation of the protein N-terminus (−89.03 Da) as variable modifications and carbamidomethylation of cysteine (+57.021 Da) and the addition of the TMT mass tag (+229.163 Da) to peptide N-termini and lysine as fixed modifications. Searches were performed with full tryptic digestion and a maximum of two missed cleavages were allowed. The reverse database search option was enabled, and all data was filtered to satisfy false discovery rate (FDR) of 5%. The mass spectrometry proteomics data have been deposited to the ProteomeXchange Consortium via the PRIDE partner repository (http://www.ebi.ac.uk/pride/archive/projects/PXD026289).

## Surface biotinylations

All solutions were pre-chilled to 4°C and all steps were carried out on ice to prevent internalisation. Fresh membrane impermeable Sulpho NHS-SS-Biotin (21331, Thermo Fisher Scientific) was dissolved in PBS at a final concentration of 0.2 mg/ml. Neurons were washed twice in PBS before being incubated with biotin for 15 mins at 4°C. The cells were then washed in PBS before being quenched in quenching buffer (50 mM Triz, 100 mM NaCl, pH 7.5) for 10 min at 4°C. The cells were lysed in 2% Triton-X-100 (X100, Sigma) plus protease inhibitor cocktail tablets (A32953, Thermo Fisher Scientific) in PBS and a BCA assay (23225, Thermo Fisher Scientific) carried out to determine protein

concentration. Equal protein amounts of lysate were incubated with streptavidin beads (17-5113-01, GE Healthcare) for 1 hr at 4°C before being washed and analysed by western blotting.

## Immunoprecipitation and western blot analysis

For immunoprecipitation experiments, cells were lysed in Tris-based immunoprecipitation buffer (50 mM Tris-HCl, pH 7.4, 0.5% NP-40, and Roche protease inhibitor cocktail in ddH$_2$O) before being subjected to GFP/RFP-Trap beads (gta-20, rta-20, ChromoTek). For whole cell levels, cells were lysed in 1% Triton-X-100 plus Roche protease inhibitor cocktail in phosphate buffered saline (PBS). A BCA assay was used to determine the protein concentration.

Proteins were resolved on NuPAGE 4–12% precast gels (NP0336BOX, Invitrogen) and then transferred onto polyvinylidene fluoride (PVDF) membranes (10600029, GE Healthcare), before being blocked in 5% milk and incubated with primary antibody O/N at 4°C. The membrane was washed in Tris-buffered saline plus 0.1% Tween (TBS-T) before being incubated with Alexa Fluor secondary antibodies (680 and 800, Invitrogen). After washing in TBS-T the protein bands were visualised using an Odyssey infrared scanning system (LI-COR Biosciences). For measuring total protein abundance all data was normalised to the protein loading control β–actin. For both total and surface protein abundance, the data was expressed as a percentage of the control treatment.

## ITC

The rat SNX27 PDZ domain and human VPS26A proteins were purified as described previously (*Gallon et al., 2014*; *McMillan et al., 2016*; *Chan et al., 2016*). Proteins were gel filtered into ITC buffer (50 mM Tris, pH 8, and 100 mM NaCl) using a Superose 200 column. The LRFN2 peptides were purchased from Genscript (USA) and ITC experiments were performed on a MicroCal iTC200 instrument in ITC buffer. Peptides at a concentration of 1 mM were titrated into 40 µM SNX27 PDZ domain solutions at 25°C (supplemented with 40 µM hVPS26A proteins when required). Data were processed using ORIGIN to extract the thermodynamic parameters $\Delta H$, $K_a(1/K_d)$ and the stoichiometry n. $\Delta G$ and $\Delta S$ were derived from the relationships: $\Delta G = -RTlnK_a$ and $\Delta G = \Delta H - T\Delta S$.

## Immunofluorescence staining

For total protein expression, neurons were fixed in 4% (vol/vol) paraformaldehyde in PBS for 15 min at RT before being quenched in 100 mM glycine. Neurons were permeabilised and blocked in 0.1% Triton X-100 plus 2% BSA (05482, Sigma) for 15 min at RT followed by incubation for 1 hr at RT in the indicated primary antibodies. The neurons were then incubated with the appropriate Alexa Fluor secondary antibodies (488, 568 and 647; Invitrogen) for 1 hr before being mounted on coverslips with Fluoromount-G (00–4958–02; eBioscience).

For surface expression of AMPA receptors, hippocampal neurons were incubated with primary antibodies that recognise the N-terminal epitope for 15 min at 37°C. For surface expression of transduced mCherry-LRFN2, neurons were incubated with a mCherry antibody. Neurons were then washed in PBS before being fixed in 4% paraformaldehyde (PFA, 28908, Thermo Fisher Scientific) in PBS for 15 min at RT and quenched in 100 mM glycine (G/0800/60, Fisher Scientific). The surface expressed AMPA receptors were detected using Alexa Fluor 568 secondary antibodies whilst the surface expressed LRFN2 was detected using a far red (647) Alexa Fluor secondary antibody to distinguish from total levels.

## Trafficking assays

For analysing LRFN2 internalisation H4 cells transduced with shRNA were transfected using Fugene six transfection reagent (E2691; Promega) with mCherry-LRFN2 for 24 hr. The cells were placed on ice, washed in ice-cold PBS and left for 5 min in CO2 independent media (18045–054, Gibco) supplemented with 1% foetal bovine serum (F7524; Sigma-Aldrich). Surface LRFN2 was labelled using the anti-rabbit mCherry antibody in CO2 independent media plus 1% foetal bovine serum for 30 min at 4°C. The cells were washed twice in ice-cold PBS and returned to the incubator in growth media for 2 hr to allow internalisation of the labelled LRFN2. The cells were washed in PBS before being fixed and permeabilised in 100% methanol for 4 min at −20°C. The cells were blocked in 1% BSA for 10 min at RT followed by incubation for 1 hr at RT with the primary antibody against LAMP2. The neurons were then incubated with the appropriate Alexa Fluor secondary antibodies (405 for the

labelled LRFN2 to distinguish from total and 647 for the LAMP2; Invitrogen) for 1 hr before being mounted on coverslips with Fluoromount-G (00–4958–02; eBioscience).

For analysing GluA2 internalisation hippocampal neurons transduced with shRNA were incubated with a GluA2 primary antibody that recognised the N-terminal epitope for 15 mins at 37°C. The neurons were then placed back into their original media in the incubator for 1 hr for the labelled GluA2 receptors to internalise. Neurons were then washed in PBS before being fixed in 4% paraformaldehyde (PFA, 28908, Thermo Fisher Scientific) in PBS for 5 min at RT. The antibody remaining on the surface was labelled with a secondary Alexa Fluor 405 antibody to block secondary binding sites for receptors that remained on the surface. The neurons were then washed in PBS, fixed again in 4% PFA for 5 min before being permeabilised and blocked in 0.1% Triton X-100 plus 2% BSA (05482, Sigma) for 10 min at RT followed by incubation for 1 hr at RT in the indicated primary antibodies (VPS35 or LAMP1). The neurons were then incubated with the appropriate Alexa Fluor secondary antibodies (568 for VPS35/LAMP1 and 647 for labelled GluA2; Invitrogen) for 1 hr before being mounted on coverslips with Fluoromount-G (00–4958–02; eBioscience).

## Image acquisition and analysis

Images were captured using a confocal laser-scanning microscope (SP5 AOBS; Leica Biosystems) attached to an inverted epifluorescence microscope (DMI6000; Thermo Fisher Scientific). A 63×, NA 1.4, oil immersion objective (Plan Apochromat BL; Leica Biosystems), and the standard SP5 system acquisition software and detector were used. All settings were kept the same within experiments. For the neuronal surface, immunofluorescence experiments single plane images were captured, whereas for the trafficking experiments looking at internalised proteins Z-stacks were compiled.

To calculate the colocalisation of LAMP2 with LRFN2 in the H4 cells, Volocity was used to calculate the Pearson's correlation. Fiji ImageJ software (NIH) was used to process all neuronal images. For colocalisation in the neurons, line traces were used across a 20 µm region of interest within the proximal dendrites. To quantify AMPA receptor surface expression after shRNA treatment, 10 neurons were imaged for each individual experiment across five independent experiments (number of experiments N = 5; total number of neurons analysed: n = 50). The maximum fluorescence intensity was measured across three 10 µm ROIs within the proximal dendrites and averaged for each neuron. To quantify the internalised GluA2 intensity after shRNA treatment, 10/12 neurons were imaged for each individual experiment across five independent experiments (number of experiments N = 5; total number of neurons analysed: n = 58). The mean intensity was measured from the cell body and proximal dendrites.

## Viral surgical procedure

Experiments were carried out in naïve male Lister Hooded rats (Envigo, UK) weighing 280–350 g at the start of experiments. Animals were housed in groups of 2–4 under a 12 hr/12 hr light/dark cycle with lights on 20:00-08:00 and were given ad libitum access to food and water. Sacrifice for ex-vivo slices occurred 2–3 hr into the dark cycle. All animal procedures were conducted in accordance with the United Kingdom Animals Scientific Procedures Act (1986) and associated guidelines. All efforts were made to minimise suffering and number of animals used.

Each animal was injected with shRNA lentiviral vectors in the dorsal hippocampus (dHPC) of one hemisphere and control vector in dHPC of the opposite hemisphere, with the experimenter blinded to viral type and viruses left and right counterbalanced. Rats were anaesthetised with isoflurane (4% induction, 2–3.5% maintenance) and secured in a stereotaxic frame with the incisor bar set 3.3 mm below the interaural line. two burr holes per hemisphere were made in the skull at the following coordinates with respect to bregma: anterior-posterior (AP) – 3.2 mm, mediolateral (ML) ± 2.4 mm and AP −3.9 mm, ML ± 2.8 mm. Virus was front loaded into a 33-gauge needle attached to a 5 µl Hamilton syringe. The needle was lowered 2.9 mm below bregma using the above AP and ML coordinates and 1 µl of virus was delivered to each site at a (therefore each hemisphere received a total of 2 µm of virus) rate of 200 nl.min$^{-1}$, with the needle left in situ for 10 min after each injection.

## Ex vivo slice preparation

Six to eight weeks following viral injection animals were anaesthetised with 4% isoflurane and decapitated. Brains were rapidly removed and placed into ice-cold sucrose cutting solution (in mM: 189

sucrose, 26 NaHCO$_3$, 10 D-glucose, 5 MgSO$_4$, 3 KCl, 1.25 NaH$_2$PO$_4$, 0.2 CaCl$_2$) saturated with 95% O$_2$/5% CO$_2$. 350-μm-thick parasagittal hippocampal slices were prepared using a vibratome (7000smz-2, Camden Instruments) and stored at room temperature in artificial cerebrospinal fluid (aCSF; in mM: 124 NaCl, 26 NaHCO$_3$, 10 D-glucose, 3 KCl, 2 CaCl2, 1.25 NaH$_2$PO$_4$, 1 MgSO$_4$) saturated with 95% O$_2$/5% CO$_2$ for $\geq$1 hr before recording. Slices were separated by hemisphere with the experimenter blind to viral type.

Electrophysiology dHPC slices were placed into a submerged recording chamber and perfused with 32–34˚C aCSF at a rate of ~2 ml.min$^{-1}$. Wide field fluorescence was used to confirm lentiviral transduction as indicated by GFP fluorescence. Two to 5 MΩ borosilicate glass electrodes (GC150-10F, Harvard Apparatus) filled with aCSF were placed into the stratum radiatum of a GFP-positive region of CA1 and a bipolar stimulating electrode (CBAPB50, FH-Co) was placed in adjacent stratum radiatum to stimulate Schaffer collaterals. Recordings were obtained using a Molecular Devices Multiclamp 700A or 700B, filtered at 4 KHz and digitised at 20 KHz using WinLTP software. Paired-pulse stimuli (50 ms inter-stimulus-interval) were delivered every 10 s using a digitimer DS2A constant voltage stimulator. Input-output curves were generated with a minimum of 3 stimuli at each stimulus intensity prior to LTP experiments. LTP induction was achieved using a single tetanus of 100 stimuli delivered at 100 Hz. Where possible two experiments per hemisphere per animal were made. Data were acquired using WinLTP (*Anderson and Collingridge, 2001*).

## Statistical analysis

All statistical analyses were carried out using GraphPad Prism 7. For biochemical data, a D'Agostino and Pearson normality test was performed. For data that was normally distributed a parametric non-paired t-test was used whereas for data not normally distributed a non-parametric Mann-Whitney U test was used. For electrophysiological data, a Shapiro-Wilk normality test was performed, all data were normally distributed. A two-way ANOVA or unpaired t-test was used to assess differences in basal transmission. LTP was assessed by a paired t-test of raw fEPSP amplitudes. For all analysis mean and standard error were calculated with *p $\leq$ 0.05, **p $\leq$ 0.01, ***p $\leq$ 0.001, ****p $\leq$ 0.0001 considered significant.

# Acknowledgements

We thank Dr Martin Playford (NIH, U.S.A) for the gift of the rabbit anti-SNX27 antibody, Dr Helen Scott and Professor James Uney for the H4 cells and the Wolfson Bioimaging Facility at the University of Bristol for their support.

# Additional information

### Funding

| Funder | Grant reference number | Author |
| --- | --- | --- |
| Medical Research Council | MR/L007363/1 | Peter J Cullen |
| Medical Research Council | MR/P018807/1 | Peter J Cullen |
| Wellcome Trust | 104568/Z/14/2 | Peter J Cullen |
| Lister Institute of Preventive Medicine | | Peter J Cullen |
| National Health and Medical Research Council | APP1136021 | Brett M Collins |
| National Health and Medical Research Council | APP1099114 | Brett M Collins |
| Biotechnology and Biological Sciences Research Council | BB/R00787X/1 | Jeremy M Henley Kevin A Wilkinson |
| Royal Society | RSRP\R1\211004 | Peter J Cullen |
| Wellcome Trust | 220260/Z/20/Z | Peter J Cullen |

The funders had no role in study design, data collection and interpretation, or the decision to submit the work for publication.

## Author contributions

Kirsty J McMillan, Conceptualization, Formal analysis, Investigation, Writing - original draft, Writing - review and editing; Paul J Banks, Conceptualization, Investigation, Writing - review and editing; Francesca LN Hellel, Ruth E Carmichael, Thomas Clairfeuille, Ashley J Evans, Investigation, Writing - review and editing; Kate J Heesom, Philip Lewis, Formal analysis, Writing - review and editing; Brett M Collins, Zafar I Bashir, Jeremy M Henley, Peter J Cullen, Conceptualization, Supervision, Funding acquisition, Writing - review and editing; Kevin A Wilkinson, Conceptualization, Supervision, Investigation, Writing - review and editing

## Author ORCIDs

Kirsty J McMillan (iD) https://orcid.org/0000-0002-4619-3309
Ruth E Carmichael (iD) http://orcid.org/0000-0003-2665-2966
Ashley J Evans (iD) http://orcid.org/0000-0002-6658-2176
Kate J Heesom (iD) http://orcid.org/0000-0002-5418-5392
Philip Lewis (iD) http://orcid.org/0000-0002-2868-2459
Brett M Collins (iD) http://orcid.org/0000-0002-6070-3774
Jeremy M Henley (iD) http://orcid.org/0000-0003-3589-8335
Kevin A Wilkinson (iD) https://orcid.org/0000-0002-8115-8592
Peter J Cullen (iD) https://orcid.org/0000-0002-9070-8349

## Ethics

Animal experimentation: All animal procedures were conducted in accordance with the United Kingdom Animals Scientific Procedures Act (1986) and associated guidelines. All efforts were made to minimise suffering and number of animals used.

## Decision letter and Author response

Decision letter https://doi.org/10.7554/eLife.59432.sa1
Author response https://doi.org/10.7554/eLife.59432.sa2

# Additional files

## Supplementary files

• Supplementary file 1. A neuronal SNX27 interactome reveals new cargoes for SNX27-mediated trafficking. (A) Raw data from TMT Interactome of SNX27 compared to the GFP control quantified across three independent experiments (N = 3) in DIV21 rat cortical neurons. (B) Filtered TMT interactome (212 proteins) showing only those proteins that were statistically significant (one-sample t-test and Benjamini–Hochberg false-discovery rate) and over two log fold change compared to the GFP control.

• Transparent reporting form

## Data availability

All data generated or analysed during this study are included in the manuscript and supporting files. The mass spectrometry proteomics data have been deposited to the ProteomeXchange Consortium via the PRIDE partner repository with the dataset identifier PXD026289 with the raw and filtered data also available in Supplementary File 1.

The following dataset was generated:

| Author(s) | Year | Dataset title | Dataset URL | Database and Identifier |
| --- | --- | --- | --- | --- |
| Cullen | 2021 | Sorting nexin-27 regulates AMPA | https://www.ebi.ac.uk/ | PRIDE, PXD026289 |

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

# Appendix 1

**Appendix 1—key resources table**

| Reagent type (species) or resource | Designation | Source or reference | Identifiers | Additional information |
|---|---|---|---|---|
| Cell line (*H. sapiens*) | Human kidney cells (Hek293T) | AATC | CRL-3216 RRID:CVCL_0063 | Authentication was from AATC. We did not independently authenticate the cell line. |
| Cell line (*M. auratus*) | Baby Hamster Kidney cells (BHK-21) | AATC | [C-13] CCL-10 RRID:CVCL_1915 | Authentication was from AATC. We did not independently authenticate the cell line. |
| Cell line (*H. sapiens*) | Neuroglioma (H4) | A gift from Dr Helen Scott and Professor James Uney | | Authenticated (STR profiling) and mycoplasma tested (absent) (Eurofins) |
| Biological sample (*R. norvegicus*) | Primary hippocampal and cortical neurons | University of Bristol Animal Services Unit | | |
| Antibody | '(mouse monoclonal)' β–actin | Sigma-Aldrich | A1978 RRID:AB_476692 | WB '(1:1000)' |
| Antibody | '(mouse monoclonal)' EEA1 | BD Biosciences | 610457 RRID:AB_397830 | IF '(1:250)' |
| Antibody | '(mouse monoclonal)' FLAG | Sigma-Aldrich | F1804 RRID:AB_262044 | WB '(1:1000)' |
| Antibody | '(mouse monoclonal)' GFP | Roche | 11814460001 RRID:AB_390913 | WB '(1:1000)' |
| Antibody | '(rabbit polyclonal)' GluA1 | Merck Millipore | ab1504 RRID:AB_2113602 | WB '(1:1000)' |
| Antibody | '(mouse monoclonal)' GluA1 | Merck Millipore | MAB2263 RRID:AB_11212678 | IF '(1:100)' |
| Antibody | '(mouse monoclonal)' GluA2 | Merck Millipore | MAB397 RRID:AB_2113875 | WB '(1:1000)' IF '(1:70)' |
| Antibody | '(rabbit polyclonal)' mCherry | Abcam | ab167453 RRID:AB_2571870 | WB '(1:1000)' IF '(1:100)' |
| Antibody | '(rabbit polyclonal)' LRFN2 | Atlas | HPA07660 | WB '(1:500)' |
| Antibody | '(mouse monoclonal)' SNX27 | Abcam | ab77799 RRID:AB_10673818 | WB for human SNX27 '(1:500)' |
| Antibody | '(rabbit polyclonal)' SNX27 | A kind gift from Dr Martin Playford, NIH, U.S.A | | WB for rat SNX27 '(1:500)' |

*Continued on next page*

*Appendix 1—key resources table continued*

| Reagent type (species) or resource | Designation | Source or reference | Identifiers | Additional information |
|---|---|---|---|---|
| Antibody | '(rabbit polyclonal)' LAMP1 | Abcam | ab24170 RRID:AB_775978 | IF '(1:200)' |
| Antibody | '(mouse monoclonal)' LAMP2 | Hybridoma bank | H4B4 RRID:AB_2134755 | IF '(1:500)' |
| Antibody | '(mouse monoclonal)' Transferrin | Santa Cruz | Sc-65882 RRID:AB_1120670 | WB '(1:1000)' |
| Antibody | '(rabbit polyclonal)' VPS35 | Abcam | ab97545 RRID:AB_10696107 | IF '(1:200)' |
| Antibody | '(rabbit)' Alexa Fluor 405 | Invitrogen | A31556 RRID:AB_221605 | IF '(1:200–400)' |
| Antibody | '(mouse)' Alexa Fluor 488 | Invitrogen | A21202 RRID:AB_141607 | IF '(1:400)' |
| Antibody | '(rabbit)' Alexa Fluor 568 | Invitrogen | A10042 RRID:AB_2534017 | IF '(1:400)' |
| Antibody | '(mouse)' Alexa Fluor 568 | Invitrogen | A10037 RRID:AB_2534013 | IF '(1:400)' |
| Antibody | '(rabbit)' Alexa Fluor 647 | Invitrogen | A31573 RRID:AB_2536183 | IF '(1:400)' |
| Antibody | '(mouse)' Alexa Fluor 647 | Invitrogen | A31571 RRID:AB_162542 | IF '(1:400)' |
| Antibody | '(mouse)' Alexa Fluor 680 | Invitrogen | A21057 RRID:AB_141436 | WB '(1:10,000)' |
| Antibody | '(rabbit)' Alexa Fluor 800 | Invitrogen | SA535571 RRID:AB_2556775 | WB '(1:10,000)' |
| Peptide, recombinant protein | LRFN2 peptides | Genscript | | |
| Chemical compound, drug | Sulpho NHS-SS-Biotin | Thermo Fisher | 21331 | |
| Chemical compound, drug | Streptavidin beads | GE Healthcare | 17-5113-01 | |
| Chemical compound, drug | GFP/RFP-Trap beads | ChromoTek | gta-20, rta-20 | |
| Recombinant DNA reagent | pEGFP-C3-GLUA1(CT) | This paper | | available on request from Kevin Wilkinson |
| Recombinant DNA reagent | pEGFP-C3-GLUA2(CT) | This paper | | available on request from Kevin Wilkinson |

*Continued on next page*

*Appendix 1—key resources table continued*

| Reagent type (species) or resource | Designation | Source or reference | Identifiers | Additional information |
|---|---|---|---|---|
| Recombinant DNA reagent | pEGFP-C3-GLUA3(CT) | This paper | | available on request from Kevin Wilkinson |
| Recombinant DNA reagent | pEGFP-C3-GLUA4(CT) | This paper | | available on request from Kevin Wilkinson |
| Recombinant DNA reagent | pEGFP-C1-LRFN1(CT) | This paper | | available on request from Peter Cullen |
| Recombinant DNA reagent | pEGFP-C1-LRFN2(CT) | This paper | | available on request from Peter Cullen |
| Recombinant DNA reagent | pEGFP-C1-LRFN3(CT) | This paper | | available on request from Peter Cullen |
| Recombinant DNA reagent | pEGFP-C1-LRFN4(CT) | This paper | | available on request from Peter Cullen |
| Recombinant DNA reagent | pEGFP-C1-LRFN5(CT) | This paper | | available on request from Peter Cullen |
| Recombinant DNA reagent | pEGFP-C1-SLC1A3(CT) | This paper | | available on request from Peter Cullen |
| Recombinant DNA reagent | pEGFP-C1-SLC4A7(CT) | This paper | | available on request from Peter Cullen |
| Recombinant DNA reagent | pEGFP-C1-LRFN1ΔPDZbm (CT) | This paper | | available on request from Peter Cullen |
| Recombinant DNA reagent | pEGFP-C1-LRFN2ΔPDZbm (CT) | This paper | | available on request from Peter Cullen |
| Recombinant DNA reagent | pEGFP-C1-LRFN4ΔPDZbm (CT) | This paper | | available on request from Peter Cullen |
| Recombinant DNA reagent | pEGFP-C1-SLC1A3ΔPDZbm (CT) | This paper | | available on request from Peter Cullen |
| Recombinant DNA reagent | pEGFP-C1-SLC4A7ΔPDZbm (CT) | This paper | | available on request from Peter Cullen |
| Recombinant DNA reagent | pEGFP-C1-LRFN2(pE786A) (CT) | This paper | | available on request from Peter Cullen |
| Recombinant DNA reagent | pEGFP-C1-LRFN2(pV789A) (CT) | This paper | | available on request from Peter Cullen |
| Recombinant DNA reagent | pcDNA3.1-FLAG-SNX27 | This paper | | available on request from Kevin Wilkinson |
| Recombinant DNA reagent | pCMV2-SEP-GluA1 | Addgene | 64942 RRID: Addgene_64942 | *Blanco-Suarez and Hanley, 2014* |
| Recombinant DNA reagent | pcDNA3-SEP-GluA2 | Addgene | 64941 RRID: Addgene_64941 | *Ashby et al., 2004* |
| Recombinant DNA reagent | pCMV2-myc-GluA2 | | | *Leuschner and Hoch, 1999* available on request from Kevin Wilkinson |

*Continued on next page*

*Appendix 1—key resources table continued*

| Reagent type (species) or resource | Designation | Source or reference | Identifiers | Additional information |
|---|---|---|---|---|
| Recombinant DNA reagent | pmCherryC1-LRFN2(FL) | This paper | | available on request from Peter Cullen |
| Recombinant DNA reagent | pmCherryC1-LRFN2(FN/TM/CT) | This paper | | available on request from Peter Cullen |
| Recombinant DNA reagent | pmCherryC1-LRFN2(TM/CT) | This paper | | available on request from Peter Cullen |
| Recombinant DNA reagent | pXLG3-GFP-H1-SNX27 shRNA (Rat) (+/- WPRE) | *Binda et al., 2019* | | Rat target sequence 5'-aagaacagcaccacagaccaa-3' available on request from Kevin Wilkinson |
| Recombinant DNA reagent | pXLG3-GFP-H1-SNX27 shRNA (Human) (+/- WPRE) | This paper | | Human target sequence 5'-aagaacagtactacagaccaa-3' available on request from Kevin Wilkinson |
| Recombinant DNA reagent | pXLG3-GFP-H1-LRFN2 shRNA (+/- WPRE) | This paper | | target sequence 5'-acgacgaggtactgattta-3' available on request from Peter Cullen |
| Recombinant DNA reagent | pXLG3-GFP-H1-control shRNA (+/-WPRE) | *Binda et al., 2019* | | non-targeting sequence 5'-aattctccgaacgtgtcac-3' available on request from Kevin Wilkinson |
| Recombinant DNA reagent | pSinRep5-GFP/ pSinRep5-GFP-SNX27 | This paper | | available on request from Kevin Wilkinson |
| Recombinant DNA reagent | pSinrep-mcherry/ pSinRep5-mCherry-LRFN2 | This paper | | available on request from Peter Cullen |

