## [Decision Letter]

**Acceptance summary:**

This study uses proteomics to identify interactors of Sorting Nexin-27 (SNX27) a known regulator of AMPA receptor trafficking. It is convincingly shown that one of the identified interactors, the integral membrane LFRN2 protein, acts as an adaptor linking SNX27 to AMPA receptors. The study also confirms previously published data showing SNX27- and LRFN2-dependent regulation of AMPA-receptors in a range of neurological conditions. Overall, these are convincing experiments revealing a novel and important interaction contributing to the control of AMPA-receptor levels and function.

**Decision letter after peer review:**

[Editors’ note: the authors submitted for reconsideration following the decision after peer review. What follows is the decision letter after the first round of review.]

Thank you for submitting your work entitled "Sorting nexin-27 regulates AMPA receptor trafficking through the synaptic adhesion protein LRFN2" for consideration by *eLife*. Your article has been reviewed by 3 peer reviewers, one of whom is a member of our Board of Reviewing Editors, and the evaluation has been overseen by a Senior Editor. The reviewers have opted to remain anonymous.

Our decision has been reached after consultation between the reviewers. Based on these discussions and the individual reviews below, we regret to inform you that your work will not be considered further for publication in *eLife*.

Two of the three reviewers felt strongly that the discovery of the adaptor function for LFRN2, though convincing, didn't make a strong enough advance for *ELife*. As you will see, shortcomings noted in the reviews are previous findings demonstrating the functional significance and questions regarding details of how LFRN2 might control trafficking and be regulated. We are sorry to have reached this decision, we thank you for your patience, and we hope that the points made in the review will be useful to you as you plan your next steps.*Reviewer #1:*

This study investigates SNX27-dependent regulation of AMPA-receptors. Evidence is provided arguing against direct SNX27/AMPA-receptor interactions. Rather, the authors discover that the integral membrane LFRN2 protein mediates indirect association of SNX27 with AMPA-receptor subunits via a cytoplasmic, C-terminal, PDZ-ligand and a lumenal domain that contains LRR, Ig and fibronectin type III domains. The PDZ interaction (SNX27/LFRN2) is shown to be direct while the exoplasmic interaction (LFRN2/GluA2) is demonstrated by co-transfection/co-IP/blot. The physiological significance of LFRN2 on AMPA-receptor function is confirmed by knockdown in both isolated cells and in animals. Overall, these are convincing experiments revealing a novel and important interaction contributing to the control of AMPA-receptor levels and function. I have only minor issues that I believe the authors could better explain.

1. Although the immuno-isolation techniques employed here might favor binary complexes, it would be expected that indirect interactions would persist to some degree. Why then is there no evidence for the ternary complex SNX27/LFRN2/GluA2 in the data?

2. The failure of LFRN2 knockdown to affect GluA1 surface levels should be explained. Especially in light of the evidence that LFRN2 binds both GluA1 and GluA2 and that, if my understanding is correct, GluA1-containing surface receptors also contain GluA2.

3. Can the stoichiometry between LFRN2 and AMPA-receptor be estimated?

*Reviewer #2 :*

This study uses proteomics to identify interactors of GFP-SNX27 in neurons using immunoprecipitation with anti-GFP nanobodies. The authors identify 212 proteins as a high-confidence SNX27 interactome. The authors follow up on LRFN2, a PDZ binding motif-containing protein, and test its role in maintaining GluA1 and GluA2 expression in neurons and AMPA receptor activity in the hippocampus. The study provides a useful dataset of proteins that immunoprecipitate with SNX27, provides support that LRFN2 interacts with SNX27, and shows that SNX27 and LRFN2 affect AMPA receptor expression and activity.

1. The main concern I have is about the specific model proposed – that SNX27 is required for LRFN2 recycling (and not for AMPA recycling?) and that LRFN2 on the surface is required for AMPAR stability on the surface. But neither aspect of the model is tested in this manuscript. To present this model, the authors need to test LRFN2 and AMPAR recycling directly using one of many established assays, and also test AMPAR destabilization on the surface, by either directly measuring mobility or following surface AMPAR expression over time after LRFN2 depletion.

AMPA receptor destabilization on the surface and recycling has been heavily studied, and the mechanisms could be more complex and redundant than the authors describe. For example LRFN1 depletion also reduces AMPA and NMDA receptors on the surface (Morimura et al). LRFN2 and AMPA receptor subunits can bind multiple PDZ proteins including PSD-95, PSD-93, and SAP97. NMDA receptors bind both SNX27 and LRFN2, and they both might act together in its surface expression. The authors need to discuss a model that integrates existing data and this complexity better.

2. LRFN2 PDZ domain is required for ER export and localization (Seabold et al., 2012). This makes Figure 3 difficult to interpret the way authors do. The loss of total and surface LRFN2 could be due to block in ER export and degradation. This mislocalization could also lead to loss of IP of the PDZ mutants.

3. Figure 4D shows that LRFN2 IP is better with GluA1 than GluA2. But Figure 4F shows better GluA2 IP. Is this a function of the myc vs. SEP tag? Also, the reduction in interaction is not obvious with GluA1 in Figure 4F. Repeating these, quantitating the levels and doing statistics, or using another quantitative assay to measure interaction, is necessary to make this conclusion.

4. The antibody staining in Figure 4 is not well described. When staining for surface expression of GluA1, GluA2 and LRFN2, were all antibodies added before fixation? Are the LRFN antibodies in the same fluorescence spectrum as mCherry, in which case we will not be able to differentiate surface from total? Is there a control surface protein or membrane staining that the authors can use?

5. It is difficult to definitively prove that two proteins do not interact. Many interactions that are accepted now cannot be easily detected using IP. Based on sequence analysis in Clairfeuille et al., GluA1 is expected to weakly bind SNX27. Considering that this is a key conclusion of the study, can the authors measure affinity of the PDZbm of GluA1 and GluA2 to SNX27 by ITC?

*Reviewer #3:*

McMillan et al., challenge the current view that SNX27 directly interacts with AMPA receptors. They convincingly show that LRFN2 acts as an adaptor linking AMPA receptors to SNX27. They went on to confirm other previously published data on the function of SNX27 and LRFN2 in a range of neurological conditions.

Most of the experiments and data presented are convincing. Most of the interaction data shown in western blots is not quantified. However, the quantification should be provided.

The role of SNX27 in AMPA receptor trafficking and neurological consequences has been addressed in:

Wang et al., Nat. Med. 2013 (studying SNX27-/- mice)

Hussain et al., PNAS 2014,

Loo et al., Nat. Comm. 2014

Damseh et al., Neurogenetics 2015 (the senior corresponding author of this manuscript is a co-author on the 2015 published study)

The role of LRFN2 in AMPA receptor trafficking and neurological consequences has been addressed in:

Morimura et al., Nat. Comm. 2017

This not a complete literature survey as LRFN proteins also go by the name of SALM.

The authors should explain better what we learn from their study besides that SNX27 does not bind directly AMPA receptors but does it through an adaptor, LRFN2.

The pathophysiological consequences of loss of function of SNX27 and LRFN2 have already been described.

[Editors’ note: further revisions were suggested prior to acceptance, as described below.]

Thank you for submitting your article "Sorting nexin-27 regulates AMPA receptor trafficking through the synaptic adhesion protein LRFN2" for consideration by *eLife*. Your article has been reviewed by 2 peer reviewers, and the evaluation has been overseen by a Reviewing Editor and Suzanne Pfeffer as the Senior Editor. The reviewers have opted to remain anonymous.

Essential Revisions:

Thanks for your careful consideration of the reviewers comments. Your study has convincing experiments revealing a novel and important interaction contributing to the control of AMPA-receptor surface levels and function. Listed verbatim below are a handful of residual issues raised by the review that can be addressed by modifications of the text.

*Reviewer #2:*

The revised manuscript includes data to address some of the concerns raised in the previous submission. The manuscript includes data that suppressing SNX27 increases LRFN2 colocalization with LAMP2, suggesting that SNX27 protects against lysosomal degradation of LRFN2, and that SNX27 and LRFN2 suppression causes increased localization of GluA2 in intracellular compartments. The manuscript also includes transferrin receptor as a control surface protein for SNX27 suppression. The concern about ER export block was partly addressed by showing that the localization of a PDZ mutant looks the same as that of the wild type. The authors have also added new discussion on AMPA receptor trafficking literature and expanded the model.

Overall, the proteomic, biochemical, and physiological data are convincing and interesting. The manuscript shows that SNX27 binds LRFN2, that SNX27 regulates LRFN2 trafficking, that SNX27 and LRFN2 regulate AMPA receptor surface levels, and that. The trafficking data support the model that SNX27 binds LRFN2 and regulates LRFN2 surface levels, and that these proteins play a role in regulating AMPA receptor surface levels. Although the exact model they propose for AMPA receptor trafficking still needs to be supported better, the manuscript is valuable in that it implicates LRFN2 as a factor that regulates AMPA receptor surface levels and thereby modulates neuronal function.

My comments are for clarity and can be addressed by revising the text. Mainly, I urge the authors to be specific in stating what the experiments test and in the conclusions drawn from the experiments, particularly for the trafficking experiments.

For example Figure 6 is titled "Suppression of SNX27 or LRFN2 increases the retention time of AMPA receptors in the endo-lysosomal system". As far as I can make out, the authors do not quantitate retention time but show only that there is more AMPA receptors inside the cell after 1 hour of internalization.

Figure 3 tests internalization and degradation (loss of surface levels and the new data that there is more colocalization of LRFN with LAMP2 after SNX27 suppression) but the conclusion is focused on recycling.

Figure 4 is titled "LRFNs interact with AMPA receptors and regulate their membrane trafficking", but membrane trafficking is not tested at all.

Figure 5 titled 'SNX27 and LRFN2 suppression affects AMPA receptor expression", do they mean surface levels or protein expression?

In Figure 6, how was the internal pool specifically quantitated (as opposed to measuring total fluorescence)? Were the neurons treated, e.g., acid washed, to remove cell surface antibodies? There is very little staining on the surface.

Does Figure 7F show the same data that are represented in B and D? If so, it is difficult to interpret because the control curves in B and D look different.

---

## [Author Response]

[Editors’ note: The authors appealed the original decision. What follows is the authors’ response to the first round of review.]

Two of the three reviewers felt strongly that the discovery of the adaptor function for LFRN2, though convincing, didn't make a strong enough advance for ELife. As you will see, shortcomings noted in the reviews are previous findings demonstrating the functional significance and questions regarding details of how LFRN2 might control trafficking and be regulated. We are sorry to have reached this decision, we thank you for your patience, and we hope that the points made in the review will be useful to you as you plan your next steps.Reviewer #1:This study investigates SNX27-dependent regulation of AMPA-receptors. Evidence is provided arguing against direct SNX27/AMPA-receptor interactions. Rather, the authors discover that the integral membrane LFRN2 protein mediates indirect association of SNX27 with AMPA-receptor subunits via a cytoplasmic, C-terminal, PDZ-ligand and a lumenal domain that contains LRR, Ig and fibronectin type III domains. The PDZ interaction (SNX27/LFRN2) is shown to be direct while the exoplasmic interaction (LFRN2/GluA2) is demonstrated by co-transfection/co-IP/blot. The physiological significance of LFRN2 on AMPA-receptor function is confirmed by knockdown in both isolated cells and in animals. Overall, these are convincing experiments revealing a novel and important interaction contributing to the control of AMPA-receptor levels and function. I have only minor issues that I believe the authors could better explain.

We sincerely thank this reviewer for expressing their positive and concise summary of our manuscript.

1. Although the immuno-isolation techniques employed here might favor binary complexes, it would be expected that indirect interactions would persist to some degree. Why then is there no evidence for the ternary complex SNX27/LFRN2/GluA2 in the data?

We thank the reviewer for this comment. The isolation of a ternary complex is dependent on the affinity of individual interactions and the amount of the steady-state complex. We suspect therefore that the lack of a quantified SNX27/LRFN2/GluA2 complex results from the low μM affinity of individual interactions and the relatively low steady-state levels of the ternary complex. When combined, these parameters work against the isolation of sufficient ternary complex for detection by western analysis.

2. The failure of LFRN2 knockdown to affect GluA1 surface levels should be explained. Especially in light of the evidence that LFRN2 binds both GluA1 and GluA2 and that, if my understanding is correct, GluA1-containing surface receptors also contain GluA2.

This is an interesting point. While the reviewer is correct that GluA1/GluA2-containing AMPA receptors represent a major population, GluA1-lacking receptors (GluA2/GluA3-containing AMPA receptors) constitute the other major heteromeric combination^1^. Indeed, differential trafficking of GluA1-containing and GluA1-lacking receptors is observed during models of synaptic plasticity – for example, the synaptic expression of GluA1-containing AMPA receptors is enhanced during LTP, while LTD is best characterised by the removal of GluA2-containing receptors ^2,3^. Thus, different effects on GluA1 versus GluA2 are not unexpected from the literature.

Since LRFN2 interacts with both GluA1- and GluA2-containing AMPA receptors it may, at first glance, seem surprising that LRFN2 loss primarily affects GluA2. However, AMPA receptor trafficking is an extraordinarily plastic process, and it remains possible that pathways exist that can act to maintain surface expression of GluA1-containing receptors (*e.g.* GluA1/GluA2 receptors) upon loss of LRFN2. However, our data suggest that for GluA1-lacking receptors this is not the case, and thus we conclude that LRFN2 plays an essential role in maintaining surface expression of this population of receptors. Consequently, we observe a reduction in GluA2 surface expression upon LRFN2 knockdown, but no effect on GluA1. Furthermore, there is the possibility that there are some redundancies between the LRFN family. We have shown that LRFN1 and LRFN4 can also interact with GluA1 and GluA2 subunits so it could be possible that other LRFN proteins maintain surface expression of GluA1-containing receptors upon loss of LRFN2. We have edited the discussion (lines: 679-687, 715-736) and the model in Figure 8 to clarify these points in our revised manuscript.

3. Can the stoichiometry between LFRN2 and AMPA-receptor be estimated?

This is another interesting question, and investigation of the subunit stoichiometry of AMPA receptors with their transmembrane regulators is an active field of research. However, while this represents an interesting question for future studies, the primary focus of our study is to establish the link between LRFN2 and SNX27-dependent AMPA receptor trafficking. As a result, we feel detailed investigation on the LRFN2-AMPA receptor stoichiometry is beyond the scope of the current study.

Reviewer #2 :This study uses proteomics to identify interactors of GFP-SNX27 in neurons using immunoprecipitation with anti-GFP nanobodies. The authors identify 212 proteins as a high-confidence SNX27 interactome. The authors follow up on LRFN2, a PDZ binding motif-containing protein, and test its role in maintaining GluA1 and GluA2 expression in neurons and AMPA receptor activity in the hippocampus. The study provides a useful dataset of proteins that immunoprecipitate with SNX27, provides support that LRFN2 interacts with SNX27, and shows that SNX27 and LRFN2 affect AMPA receptor expression and activity.

We thank the reviewer for this summary.

1. The main concern I have is about the specific model proposed – that SNX27 is required for LRFN2 recycling (and not for AMPA recycling?) and that LRFN2 on the surface is required for AMPAR stability on the surface. But neither aspect of the model is tested in this manuscript. To present this model, the authors need to test LRFN2 and AMPAR recycling directly using one of many established assays, and also test AMPAR destabilization on the surface, by either directly measuring mobility or following surface AMPAR expression over time after LRFN2 depletion.

We apologise that our model was not clearly stated. Our data do not contradict that AMPA receptor recycling requires SNX27, indeed we have shown that surface levels of AMPA receptors are decreased after SNX27 suppression (Figure 3D and Figure 5A). Rather we argue that recycling is not principally controlled by direct SNX27 binding to AMPA receptors (the present dogma). Of course, we cannot conclude that under certain circumstances SNX27 may directly bind to AMPA receptors. We do propose however, that SNX27 can control AMPA receptor expression and activity through its regulation of the ‘bridging’ adaptor LRFN2. We have revised the manuscript and edited the model in an attempt to make this clearer (lines: 40-44, 128-137, 218-233, 255-262, 695-707, 796-799).

To provide additional support of the proposed model, we have carried out new experiments to further investigate the trafficking of LRFN2 and AMPA receptors. Unfortunately, we did not have a reliable antibody for detecting endogenous LRFN2 by immunofluorescence. Therefore, to investigate LRFN2 recycling we transfected the neuroglioma H4 cells with a construct encoding for N-terminally mCherry tagged full length LRFN2. Using an anti-mCherry antibody to label the exofacial mCherry epitope, cell surface uptake assay have revealed that in SNX27 suppressed cells, the recycling of LRFN2 is perturbed as shown by an increased colocalization of LRFN2 with the lysosomal marker LAMP2 (see Figure 3G; lines: 405-415). In combination with biochemical data from primary neurons where we have analysing total and surface expression of LRFN2 after SNX27 suppression, these data demonstrate that SNX27 is required for LRFN2 trafficking to the cell surface and for preventing LRFN2 missorting into the degradative lysosomal compartment.

To test which step of AMPA receptor trafficking SNX27 and LRFN2 suppression affects we labelled surface GLUA2 receptors and then looked at their internalisation in rat hippocampal neurons. In both SNX27 and LRFN2 suppressed neurons we found an increase in the intensity of internalised GLUA2 in the endo-lysosomal system (co-labelled with EEA1 and LAMP1). This suggests that the recycling of GLUA2 is perturbed thereby increasing its retention time in the endosomal system. We have updated the manuscript and our model to integrate these new data (Figures 6 and 8; lines: 515-531).

Our reference to LRFN2 regulating the stability of cell surface AMPA receptors comes from the existing literature where LRFNs are considered to cluster receptors at the synaptic surface. We agree with the reviewer that this has not been tested in our model and so we have revised the manuscript and the model to make this clearer (Figure 8; lines: 762-771).

AMPA receptor destabilization on the surface and recycling has been heavily studied, and the mechanisms could be more complex and redundant than the authors describe. For example LRFN1 depletion also reduces AMPA and NMDA receptors on the surface (Morimura et al). LRFN2 and AMPA receptor subunits can bind multiple PDZ proteins including PSD-95, PSD-93, and SAP97. NMDA receptors bind both SNX27 and LRFN2, and they both might act together in its surface expression. The authors need to discuss a model that integrates existing data and this complexity better.

Our study shows that LRFN2 can associate with AMPA receptors and mediate their SNX27-dependent trafficing but we completely agree with the reviewer that the molecular mechanisms of AMPA and NMDA receptor trafficking, both at the cell surface and within the endosomal network, are highly complex and tightly regulated. To clarify this point we have added further discussion to the revised manuscript (lines: 753-778) and have edited the model in Figure 8 to further integrate these complexities.

2. LRFN2 PDZ domain is required for ER export and localization (Seabold et al., 2012). This makes Figure 3 difficult to interpret the way authors do. The loss of total and surface LRFN2 could be due to block in ER export and degradation. This mislocalization could also lead to loss of IP of the PDZ mutants.

In the experiments shown in Figure 3 we have suppressed SNX27 and measured the total and surface levels of endogenous LRFN2. Therefore, the PDZ binding motif present in the LRFN2 tail is intact. These experiments do not target LRFN2 directly nor are any PDZ binding motif mutants used, hence there should be no block in ER export of the endogenous protein. The cell-based IPs with wild type LRFN’s and corresponding PDZ mutants are all derived from expression constructs encoding for just the cytoplasmic tail domains, they do not contain the transmembrane domain. These constructs are all therefore expressed in the cytoplasm and hence ‘mis-localisation’ in the context of being trapped in the ER is not an issue.

To address these concerns further however we transduced rat hippocampal neurons with sindbis viruses expressing LRFN2 (with and without the PDZbm) and stained with the ER marker Calnexin (Author response image 1). We could see no evidence of increased colocalization of the LRFN2ΔPDZbm construct in the ER compared to the LRFN2 construct.

**Author response image 1. sa2fig1:** 

3. Figure 4D shows that LRFN2 IP is better with GluA1 than GluA2. But Figure 4F shows better GluA2 IP. Is this a function of the myc vs. SEP tag? Also, the reduction in interaction is not obvious with GluA1 in Figure 4F. Repeating these, quantitating the levels and doing statistics, or using another quantitative assay to measure interaction, is necessary to make this conclusion.

We thank the reviewer for highlighting this point. We have added the quantification and statistical analysis for Figure 4F showing the reduction in interaction of the LRFN2 mutants with GLUA1 and GLUA2 (Figure 4F; lines 448-460). We did find that the myc-GluA2 expressed better in HEK293T cells than the GFP-GluA2; which can be seen by the low GFP expression of the GluA2 construct in Figure 4D.

4. The antibody staining in Figure 4 is not well described. When staining for surface expression of GluA1, GluA2 and LRFN2, were all antibodies added before fixation? Are the LRFN antibodies in the same fluorescence spectrum as mCherry, in which case we will not be able to differentiate surface from total? Is there a control surface protein or membrane staining that the authors can use?

We apologise that we did not make this clearer. We have added text to both the figure legend and the methods (lines: 430-436, 469-478, 1028-1036). The antibodies to GluA1 or GluA2, and to mCherry, were added prior to fixation to only allow staining of surface expressed proteins. The surface mCherry staining, representing surface expressed LRFN2, was then detected using a far-red labelled secondary antibody, allowing distinction between far-red (surface LRFN2) and red (total LRFN2) signals.

5. It is difficult to definitively prove that two proteins do not interact. Many interactions that are accepted now cannot be easily detected using IP. Based on sequence analysis in Clairfeuille et al., GluA1 is expected to weakly bind SNX27. Considering that this is a key conclusion of the study, can the authors measure affinity of the PDZbm of GluA1 and GluA2 to SNX27 by ITC?

In Supplementary Figure 3 of Clairfeuille et al., ITC quantification established that the PDZ domain of SNX27 does not show significant binding to GluA1 or GluA2 ^4^. Moreover, in the same Figure the authors establish that binding is not enhanced by phosphorylation of serine or tyrosine residues found in GluA1 or by association with VPS26. We have now included specific reference to these data in the revised manuscript (lines: 114-121). However, we agree that it is difficult to definitively prove a negative and have revised the manuscript to address this point (lines: 257-260).

Reviewer #3:McMillan et al., challenge the current view that SNX27 directly interacts with AMPA receptors. They convincingly show that LRFN2 acts as an adaptor linking AMPA receptors to SNX27. They went on to confirm other previously published data on the function of SNX27 and LRFN2 in a range of neurological conditions. Most of the experiments and data presented are convincing. Most of the interaction data shown in western blots is not quantified. However, the quantification should be provided.

We thank the reviewer for this positive summary and have included quantification for the immunoprecipitation experiments where there is a reduction in binding in the revised manuscript (Figures 1F, 2B, 2C, 4F).

The role of SNX27 in AMPA receptor trafficking and neurological consequences has been addressed in:Wang et al., Nat. Med. 2013 (studying SNX27-/- mice)Hussain et al., PNAS 2014,Loo et al., Nat. Comm. 2014Damseh et al., Neurogenetics 2015 (the senior corresponding author of this manuscript is a co-author on the 2015 published study)

We are not disputing the evidence that SNX27 controls AMPA receptor endocytic trafficking and have included these references in the manuscript. Rather we argue that besides direct SNX27 binding to AMPA receptors (the present dogma), LRFN2 acts as a ‘bridging’ adaptor in SNX27-dependent endosomal recycling of internalized AMPA receptors. We believe that this constitutes an important discovery in our current understanding of SNX27-mediated AMPA receptor trafficking. In suggesting a revised consideration of SNX27-mediated AMPA receptor recycling through identifying LRFN2 as a component of this pathway, we strongly believe our study represents an important advance in our understanding. We have edited the manuscript to clarify these points further (lines: 40-44, 128-137, 218-233, 255-262, 695-707, 796-799).

The role of LRFN2 in AMPA receptor trafficking and neurological consequences has been addressed in: Morimura et al., Nat. Comm. 2017

Respectfully we disagree that the role of LRFN2 in AMPA receptor trafficking has been addressed. We demonstrate that LRFN2 contributes to AMPA receptor trafficking through coupling receptors to SNX27 – this has certainly not been described previously. While two studies have examined hippocampal LTP in LRFN2 knockout mice (Li et al., 2018 and, as the reviewer notes, Morimura et al., 2017, both of which are referenced in our manuscript lines: 737-752), the findings have been contradictory, with Li et al. observing reduced LTP in LRFN2 knockout mice, and Morimura et al., reporting enhanced LTP upon loss of LRFN2. Given this contradictory data, and lack of mechanistic insight into the link between LRFN2 and AMPA receptors, we believe our data is a significant addition to the field. Moreover, our molecular level investigation of the role of LRFN2, combined with ex vivo electrophysiology, not only adds insight into the role of LRFN2, but clarifies the contradictory evidence relating to the role of LRFN2 in plasticity by examining post-development suppression of LRFN2 and its consequences on AMPA receptor activity (compared to KO studies in which LRFN2 has been removed throughout development). We have edited the Discussion section of our manuscript to clarify the significance of our study further (lines: 782-799).

This not a complete literature survey as LRFN proteins also go by the name of SALM.

We apologise if this was not clear in our manuscript and have clarified this point in our revised manuscript (lines: 266, 699-700). In the references we have included many of the original SALM papers (Wang et al., 2006; Ko et al., 2006; Seabold et al., 2008; Mah et al., 2010; Nam et al., 2011; Lie et al., 2018; Lin et al., 2018; Brouwer et al., 2019). The official gene/protein name for the LRFN family is LRFN1, LRFN2 etc which is why we chose to use this nomenclature throughout the manuscript.

The authors should explain better what we learn from their study besides that SNX27 does not bind directly AMPA receptors but does it through an adaptor, LRFN2.

There is an increasing interest in the role of the endosomal system in neurodegenerative disease with endosomal swelling classed as an early feature of disease. This enlargement of endosomes indicates a trafficking defect, but the molecular mechanisms of how endosomal dysfunction leads to neurodegeneration is unclear. An increasing body of work is focusing on the role of the SNX27 which has been associated with several neuropathologies including Down’s syndrome. Many of these studies focus on the role of SNX27 in regulating AMPA receptor trafficking, leading to a dogma that places SNX27’s ability to directly bind these receptors at its mechanistic heart. In this study we define an interaction between SNX27 and a synaptic adhesion molecule LRFN2 which contributes to the control of glutamatergic transmission and activity which we believe represents an important advance in the field. LRFN2 is increasingly being linked to neurodegenerative diseases with our study providing new molecular insight into the role LRFN2 could play in these diseases through its regulation of AMPA receptor expression.

Besides the identification of the SNX27:LRFN2:AMPA receptor pathway, our unbiased proteomic analysis in primary neurons has also identified an array of new SNX27 cargo proteins (see Figure 1), many of which provide new insight into SNX27’s potential role in regulating neuronal activity. Their identification is likely to drive additional research into the underlying causes of SNX27-associated pathologies associated with neurodegeneration, epilepsy, spastic paraplegia and intellectual disability. We have edited the Discussion section of the manuscript to address these points and to highlight further the significance of our study.

The pathophysiological consequences of loss of function of SNX27 and LRFN2 have already been described.

While this is true, these neuronal pathologies have not previously been considered as functionally linked. In describing a mechanistic link (at least in the context of AMPA receptor trafficking) we provide clear evidence that these pathologies are coupled. Furthermore, the association between LRFN2 and AMPA receptors has not previously been recognised as a potential mechanism for LRFN2’s role in cognitive decline. Thus, in addition to adding molecular insight into the role of SNX27 and LRFN2 in AMPA receptor trafficking, our manuscript provides an important link between SNX27 and LRFN2-dependent cognitive decline, providing new understanding of how the co-operative action of these proteins maintains cognitive function. We have edited the manuscript to clarify these points further (lines: 782-799).

References

1. Wenthold, R. J., Petralia, R. S., Blahos, J., II & Niedzielski, A. S. Evidence for multiple AMPA receptor complexes in hippocampal CA1/CA2 neurons. *J Neurosci* 16, 1982-1989 (1996).

2. Diering, G. H. & Huganir, R. L. The AMPA Receptor Code of Synaptic Plasticity. *Neuron* 100, 314-329, doi:10.1016/j.neuron.2018.10.018 (2018).

3. Huganir, R. L. & Nicoll, R. A. AMPARs and synaptic plasticity: the last 25 years. *Neuron* 80, 704-717, doi:10.1016/j.neuron.2013.10.025 (2013).

4. Clairfeuille, T. *et al.* A molecular code for endosomal recycling of phosphorylated cargos by the SNX27-retromer complex. *Nature structural & molecular biology* 23, 921-932, doi:10.1038/nsmb.3290 (2016).

5. Gallon, M. *et al.* A unique PDZ domain and arrestin-like fold interaction reveals mechanistic details of endocytic recycling by SNX27-retromer. *Proceedings of the National Academy of Sciences of the United States of America* 111, E3604-3613, doi:10.1073/pnas.1410552111 (2014).

[Editors’ note: what follows is the authors’ response to the second round of review.]

Essential Revisions:Thanks for your careful consideration of the reviewers comments. Your study has convincing experiments revealing a novel and important interaction contributing to the control of AMPA-receptor surface levels and function. Listed verbatim below are a handful of residual issues raised by the review that can be addressed by modifications of the text. Please resubmit a revision and cover letter explaining the changes before a final decision.Reviewer #2:The revised manuscript includes data to address some of the concerns raised in the previous submission. The manuscript includes data that suppressing SNX27 increases LRFN2 colocalization with LAMP2, suggesting that SNX27 protects against lysosomal degradation of LRFN2, and that SNX27 and LRFN2 suppression causes increased localization of GluA2 in intracellular compartments. The manuscript also includes transferrin receptor as a control surface protein for SNX27 suppression. The concern about ER export block was partly addressed by showing that the localization of a PDZ mutant looks the same as that of the wild type. The authors have also added new discussion on AMPA receptor trafficking literature and expanded the model.Overall, the proteomic, biochemical, and physiological data are convincing and interesting. The manuscript shows that SNX27 binds LRFN2, that SNX27 regulates LRFN2 trafficking, that SNX27 and LRFN2 regulate AMPA receptor surface levels, and that. The trafficking data support the model that SNX27 binds LRFN2 and regulates LRFN2 surface levels, and that these proteins play a role in regulating AMPA receptor surface levels. Although the exact model they propose for AMPA receptor trafficking still needs to be supported better, the manuscript is valuable in that it implicates LRFN2 as a factor that regulates AMPA receptor surface levels and thereby modulates neuronal function.My comments are for clarity and can be addressed by revising the text. Mainly, I urge the authors to be specific in stating what the experiments test and in the conclusions drawn from the experiments, particularly for the trafficking experiments.For example Figure 6 is titled "Suppression of SNX27 or LRFN2 increases the retention time of AMPA receptors in the endo-lysosomal system". As far as I can make out, the authors do not quantitate retention time but show only that there is more AMPA receptors inside the cell after 1 hour of internalization.Figure 3 tests internalization and degradation (loss of surface levels and the new data that there is more colocalization of LRFN with LAMP2 after SNX27 suppression) but the conclusion is focused on recycling.Figure 4 is titled "LRFNs interact with AMPA receptors and regulate their membrane trafficking", but membrane trafficking is not tested at all.Figure 5 titled 'SNX27 and LRFN2 suppression affects AMPA receptor expression", do they mean surface levels or protein expression?In Figure 6, how was the internal pool specifically quantitated (as opposed to measuring total fluorescence)? Were the neurons treated, e.g., acid washed, to remove cell surface antibodies? There is very little staining on the surface.Does Figure 7F show the same data that are represented in B and D? If so, it is difficult to interpret because the control curves in B and D look different.

In response to Reviewer 2’s comments we have revised the text to improve the clarity of our results including the titles and conclusions for Figures 3, 4, 5 and 6 (lines: 418-421; 467; 539; 556-566) and thank the reviewer for these suggestions. For Figure 6 we apologise for the confusion in our methodology and have updated the text in the figure legend and results and methods sections (lines: 519-524; 556-566; 1070-1074; 1086-1088). In brief, after antibody feeding with a N-terminally tagged GluA2 antibody, the antibody remaining on the surface was labelled with a secondary Alexa Fluor 405 antibody to block secondary binding sites for receptors that remained on the surface. Internalised receptors were then detected by permbealisation followed by staining with a distinct secondary Alexa Fluor 647 antibody. As the reviewer mentioned there is very little staining on the surface indicating that we are specifically detecting internalised GluA2 for quantification. In addition, for further clarity, we have added the surface staining in Figure 6 —figure supplement 1 (562-564; 1461-1472).

For Figure 7F, the same data are represented in B and D. We have analysed the input-output for the SNX27 and LRFN2 control shRNAs and found that they were not significantly different, validating the comparison of the two shRNA groups. We have now amended the text to report the data and also state that we are cautious to over-interpret this result (lines: 596-601).